# Genome-scale metabolic modeling reveals metabolic trade-offs associated with lipid production in *Rhodotorula toruloides*

**Alīna Reķēna[1], Marina J. Pinheiro[2], Nemailla Bonturi[1], Isma Belouah[1], Eliise Tammekivi[3], Koit Herodes[3], Eduard J. Kerkhoven[4], Petri-Jaan Lahtvee[1]***

**1** Department of Chemistry and Biotechnology, Tallinn University of Technology, Tallinn, Estonia,
**2** Department of Materials and Bioprocess Engineering, University of Campinas, Campinas, Brazil,
**3** Institute of Chemistry, University of Tartu, Tartu, Estonia, **4** Department of Biology and Biological Engineering, Chalmers University of Technology, Gothenburg, Sweden

* lahtvee@taltech.ee

**Data Availability Statement:** All relevant data are within the paper, Supporting Information and on a Github repository at https://github.com/

## Abstract

*Rhodotorula toruloides* is a non-conventional, oleaginous yeast able to naturally accumulate high amounts of microbial lipids. Constraint-based modeling of *R. toruloides* has been mainly focused on the comparison of experimentally measured and model predicted growth rates, while the intracellular flux patterns have been analyzed on a rather general level. Hence, the intrinsic metabolic properties of *R. toruloides* that make lipid synthesis possible are not thoroughly understood. At the same time, the lack of diverse physiological data sets has often been the bottleneck to predict accurate fluxes. In this study, we collected detailed physiology data sets of *R. toruloides* while growing on glucose, xylose, and acetate as the sole carbon source in chemically defined medium. Regardless of the carbon source, the growth was divided into two phases from which proteomic and lipidomic data were collected. Complemental physiological parameters were collected in these two phases and altogether implemented into metabolic models. Simulated intracellular flux patterns demonstrated the role of phosphoketolase in the generation of acetyl-CoA, one of the main precursors during lipid biosynthesis, while the role of ATP citrate lyase was not confirmed. Metabolic modeling on xylose as a carbon substrate was greatly improved by the detection of chirality of D-arabinitol, which together with D-ribulose were involved in an alternative xylose assimilation pathway. Further, flux patterns pointed to metabolic trade-offs associated with NADPH allocation between nitrogen assimilation and lipid biosynthetic pathways, which was linked to large-scale differences in protein and lipid content. This work includes the first extensive multi-condition analysis of *R. toruloides* using enzyme-constrained models and quantitative proteomics. Further, more precise $k_{cat}$ values should extend the application of the newly developed enzyme-constrained models that are publicly available for future studies.

## Author summary

Transition towards a biobased, circular economy to reduce the industrial dependence on fossil-based resources requires new technologies. One of the options is to convert available

alinarekena/ecRhtoGEM. LC-MS/MS data have been deposited to the ProteomeXchange Consortium (http://proteomecentral. proteomexchange.org) via the PRIDE partner repository with the dataset identifier PXD037281.

**Funding:** This work was supported by the Estonian Research Council grants PUT1488P, PRG1101 (AR, MJP, NB, IB, PJL) and PRG1198 (ET) and NordForsk grant 103506 (AR, NB, PJL). MJP would additionally like to acknowledge Coordination for the Improvement of Higher Education Personnel (Capes), São Paulo Research Foundation (FAPESP, grant 2016/10636-8). EJK acknowledges funding by the Novo Nordisk Foundation (grant NNF10CC1016517), and the Research Council for Environment, Agricultural Sciences, and Spatial Planning (Formas) (grant 2018-00597). The funders had no role in study design, data collection and analysis, decision to publish, or preparation of the manuscript.

**Competing interests:** Authors declare no competing interests.

biomass feedstocks into valuable chemicals using microbes as biocatalysts. *Rhodotorula toruloides* is a nonpathogenic, nonconventional yeast that has recently emerged as one of the most promising yeasts for sustainable production of chemicals and fuels due to its natural ability to synthesize large amounts of lipids. However, its unique metabolic properties are not yet fully understood. We have computationally predicted metabolic fluxes in *R. toruloides* while growing in economically viable growth conditions inducing lipid accumulation and analyzed them together with absolute proteome quantification. Our holistic approach has highlighted metabolic pathways important for lipid biosynthesis and revealed metabolic trade-offs associated with NADPH allocation during lipogenesis. In addition, our work highlighted the necessity for accurate computational approaches in characterizing enzymatic kinetic properties that would improve the metabolic studies of *R. toruloides*.

## Introduction

*R. toruloides* is a red basidiomycota known for its ability to accumulate high amounts of intracellular lipids [1] and consume different carbon substrates [2,3]. It has been studied for its ability to consume complex biomass substrates, including from the lignocellulosic origin [4–6] that would make it interesting for a biorefinery concept. However, studies aimed at fundamental investigation of *R. toruloides* metabolism have been mainly conducted using a single carbon source as substrate, such as xylose [1,7–9], glucose [8–11], glycerol [7], acetate [7], L-arabinose and *p*-coumarate [9], in a chemically defined mineral medium and occasionally rich cultivation medium (YP) [3]. Secondary nutrient limitation induces lipid accumulation [12]. In nitrogen limitation, 65% of lipids of dry cell weight were reached in a batch cultivation regime [1].

Metabolic pathways producing intracellular metabolite acetyl-CoA and a cofactor NADPH in *R. toruloides* have been the main focus of metabolic studies due to their central role in lipid biosynthesis. Fatty acids, which mainly accumulate in the form of triacylglycerols (TAGs), are produced via the sequence of four enzymatic reactions that require 1 ATP and 2 NADPH molecules per 1 acetyl-CoA added to the fatty acid chain [13]. To study lipid metabolism in *R. toruloides*, previous studies have taken the systems biology approach. Genome sequencing and high-throughput multi-omics analysis facilitated the reconstruction of the metabolic networks. Based on a genome sequence of *R. toruloides* strain NP11, the first metabolic network of *R. toruloides* included its central carbon metabolism and lipid biosynthetic pathways [10]. *R. toruloides* possesses several enzymatic pathways that differ from the model yeast *Saccharomyces cerevisiae* and which specifically facilitate the generation of lipid precursors. The key differences included the synthesis of acetyl-CoA from citrate by ATP citrate lyase (ACL), synthesis of acetyl-CoA from xylulose 5-phosphate by phosphoketolase (XPK), and the conversion of S-malate into pyruvate by malic enzyme (ME) that provides for NADPH [10,14]. Proteomics analysis has suggested NADPH regeneration primarily through the pentose phosphate pathway on xylose and glucose, but the role of malic enzyme is not clearly understood [8,10]. The role of XPK in the generation of acetyl-CoA has not been acknowledged previously, whereas ACL has been demonstrated to be upregulated during lipid accumulation [10], especially in presence of xylose [8].

It has been reported that on xylose *R. toruloides* is growing 2 to 3 times slower compared to glucose [8], but the underlying mechanisms are yet to be identified. In our previous proteomics study [1], we discovered from proteomics quantification that xylulokinase, encoded in

the genome as the third step in the currently known xylose assimilation pathway, is not present in the proteomic data set, suggesting potential limitation in xylose metabolism. Later, a similar finding was reported by Jagtap et al. 2021 [3] and Kim et al. 2021 [9] using a different *R. toruloides* strain, IFO 0880. The latter proposed an alternative xylose assimilation pathway for this species.

A holistic view on metabolism can be provided by genome-scale metabolic models (GEMs). GEMs are metabolic networks reconstructed from a genome sequence of a specific organism. They contain all known biochemical reactions of the organism. GEMs allow the calculation of metabolic fluxes that represent activity of different metabolic pathways under specified conditions, e.g., an uptake of a particular carbon source. GEMs of *R. toruloides* were built based on the genome sequences of strains NP11 [15] and IFO 0880 [11]. Flux balance analysis predicted that up to 87% of NADPH was regenerated from xylose through the oxidative part of pentose phosphate pathway (oxPPP) [1]. Phosphoketolase was suggested as the main supplier of acetyl-CoA during lipogenesis in xylose-grown cells [1]. On the other hand, TCA cycle related enzymes were suggested for NADPH production on acetate-grown cells [7], demonstrating that metabolic operations can vary significantly with the carbon source uptake. Models have also been used to study metabolism during cell growth on glucose [11] and glycerol [7].

A better understanding of how different metabolic pathways contribute to lipid accumulation under different substrates would help to design better metabolic engineering strategies. GEMs can be a powerful and helpful tool in metabolic studies, if their predictive power is good. Enzyme-constrained GEMs integrate additional constraints on enzyme capacity and their total abundances (as thoroughly reviewed by Chen and Nielsen 2021 [16]). Phenomenological constraint is imposed on metabolic flux (v; mmol/gDCW/h), formulated as enzyme kinetics (**Eq 1**)

$$v \leq E \cdot k_{cat} \tag{1}$$

where E is protein abundance (mmol/gDCW) and $k_{cat}$ is the enzyme's turnover number (1/s), provided with an upper limit on individual or total protein abundances The integration of enzymatic constraints in *S. cerevisiae* significantly improved phenotype prediction [17]. The strength of proteome constraints has also been demonstrated by predicting overflow metabolism in *E. coli* [18] and metabolic shift in arginine catabolism in *L. lactis* [19]. A similar coarse-grained approach that allowed the prediction of maximal growth without constraining the model with any exchange fluxes in *S. cerevisiae* was demonstrated by applying a global thermodynamics constraint [20].

In addition to a curated annotation, the quality of the predicted fluxes depends on accuracy of physiological data, notably on the biomass composition specificity. The tuning of *R. toruloides* biomass reaction in the prior and current models improved the condition-wise specificity of predicted fluxes.

In the present study, we created condition-specific enzyme-constrained genome-scale metabolic models of *R. toruloides*, ecRhtoGEMs, and used them to predict intracellular fluxes. Flux bounds to constrain the model were obtained from bioreactor (1 L) experiments with yeast cultivation in chemically defined medium, with three carbon sources studied individually—glucose, xylose and acetate. These very detailed physiological data sets enabled us to precisely characterize metabolism at exponential growth and lipid accumulation phase. In all conditions, we performed mass spectrometry (MS) based absolute proteome quantification. Also, biomass macromolecular composition in regard to lipids and proteins was determined, including lipid profiling by gas chromatography (GC) analysis. Using this data, we generated 6 different versions of the *R. toruloides* model with enzyme constraints and biomass composition

specificity, where we were able to demonstrate trade-offs in NADPH requirements for the cells growing exponentially versus in nitrogen limitation. To our knowledge, this is the first proteomics analysis of acetate-grown *R. toruloides* cells and the first detailed GEM analysis combined with proteome analysis of acetate and glucose conditions for this strain.

## Results

### Differences in physiological parameters under glucose, xylose or acetate as a sole carbon source

Here we present production profile, specific growth rate, lipidomics and total protein measurements of batch cultivation of *R. toruloides* strain CCT 7815 growing in a chemically defined medium on three substrates as a sole carbon source—glucose (63 g/L), xylose (70 g/L) or acetate (20 g/L). *R. toruloides* CCT 7815 is a tolerant strain developed during a short-term adaptation of strain CCT 0783 (Coleção de Culturas Tropicais, Fundação André Tosello, Campinas, Brazil) in sugarcane bagasse hemicellulosic hydrolyzate, demonstrating an increased lipid production without impacting growth and substrate consumption as a result of induction of hydrolysate-tolerance- and lipid accumulation-related genes [21]. Cultures were grown at a starting molar C/N ratio of 69 (glucose/urea) and 80 (xylose- or acetate/ammonium sulfate), which will result in nitrogen limitation that is known to induce lipid accumulation [12]. Cell growth was monitored by online biomass measurements and $CO_2$ production data. Experiments were run until complete substrate depletion. Regardless of the carbon source, the results demonstrated two distinct growth phases: (i) exponential growth (exp) phase where all substrates were in excess, and (ii) nitrogen-limited (Nlim) phase, associated with nitrogen depletion (**Fig 1A**). For lipidomics, the first sample was analyzed at the end of exp phase and the second sample was analyzed at the end of Nlim phase (**Fig 1A**). For intracellular protein content analysis, biomass samples were analyzed at the late or end exp and mid-Nlim phases (**Fig 1A**). Physiological parameters are available in **S1 Table**.

The highest amount of intracellular lipids was accumulated while cells were growing on glucose, resulting in 0.48±0.04 g/gDCW, while the lipid yield was approximately 15% and 20% lower on acetate and xylose, respectively (**Fig 1B**). On glucose, lipid accumulation started later than on xylose and acetate, where up to 20% and 18% lipid yield, respectively, was reached already during the late exp phase (**S1 Fig**). In a similar study using a different *R. toruloides* strain NP11 [8], less lipids were quantified in xylose at the late exp phase, while a higher final lipid yield was reached compared to our study. On glucose, the final lipid yield was comparable with previous studies [8]. In acetate condition, the final lipid content was 0.34±0.01 g/gDCW, which was in line with previous experiments by our group measured in continuous cultivation experiments [7].

Maximum specific growth rate was the highest on glucose, 0.19±0.025 h$^{-1}$, while it was at least 2-fold lower on acetate and xylose (**Fig 1C**). Two-fold difference in maximum growth rate on xylose and glucose conditions has been reported previously [8]. Nlim growth phase, where most of the lipid accumulation occurred in all studied conditions, was characterized by significantly lower specific growth rate (**Fig 1C**), specific substrate uptake rate (**Fig 1D**) and total protein content (**Fig 1E**).

Lipid composition was similar in all studied conditions, with oleate (C18:1) as the dominant fatty acid (**Fig 1F**). During the Nlim phase, the relative amount of oleate (monounsaturated fatty acids, MUFAs) further increased, while polyunsaturated fatty acids (PUFAs)—linoleate (C18:2) and linolenate (C18:3)—decreased. Interestingly, on glucose at late exp phase the amount of PUFAs was higher than MUFAs, but it significantly changed during the Nlim phase when total lipid amount increased almost 10-fold (**Fig 1B**). Our results demonstrated that the

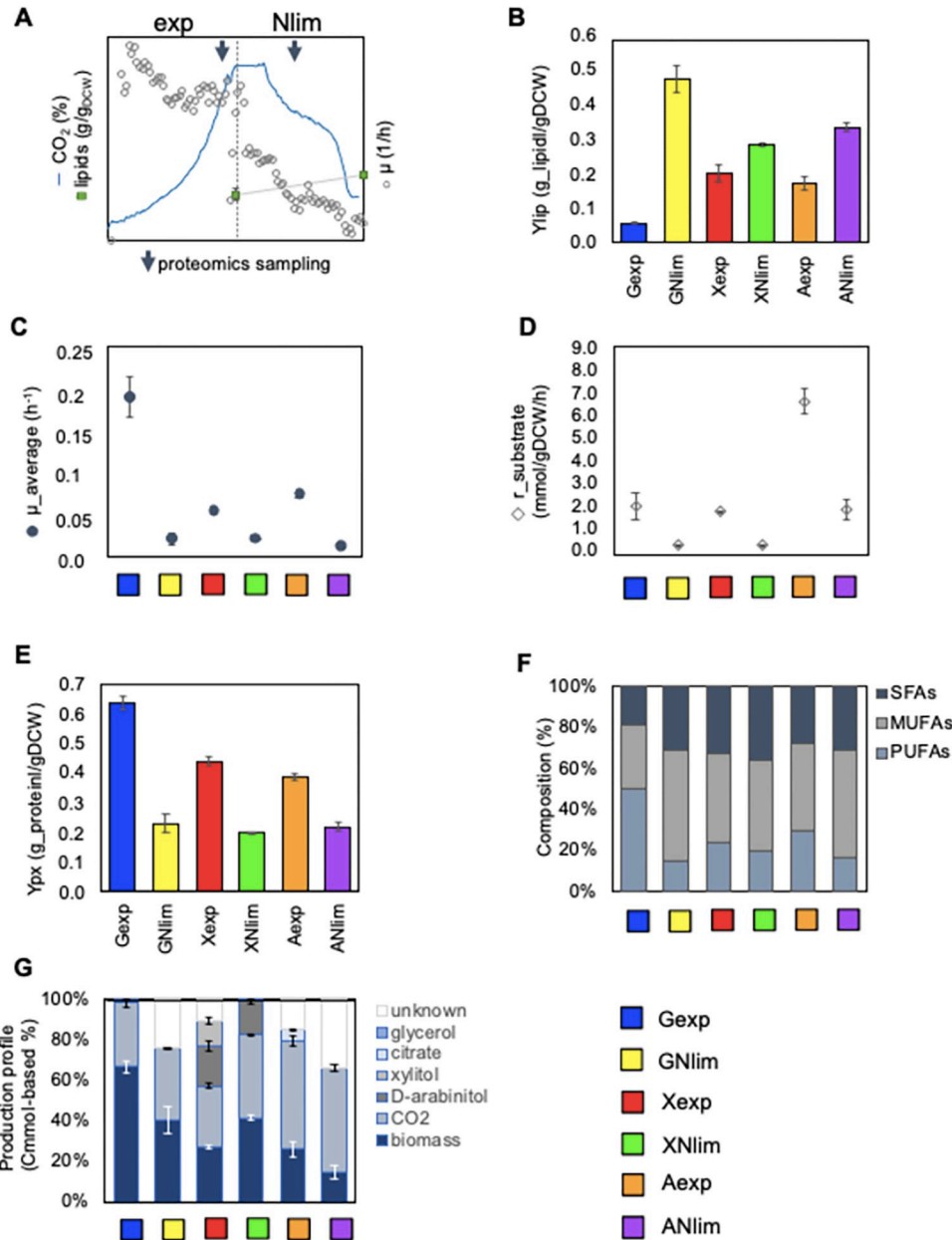

**Fig 1. Growth characterization on three carbon sources–glucose (G), xylose (X) and acetate (A)—during exponential growth (exp) and nitrogen limitation (Nlim) phases.** (A) Batch cultivation growth curve and sampling timepoints for lipidomics and proteomics on all tested carbon sources. (B) Lipid yield, Ylip (g/gDCW). (C) Average specific growth rate, μ_average (h⁻¹). (D) Substrate uptake rate, r_substrate (mmol/gDCW/h). (E) Protein content, Yp (g/gDCW). (F) Fatty acid profiles (% of total lipid). (G) Carbon balance (% of total substrate uptake). SFAs: saturated fatty acids; MUFAs: monounsaturated fatty acids; PUFAs: polyunsaturated fatty acids. Average of duplicate experiments with SD is illustrated.

degree of fatty acid saturation for C18 dynamically increased throughout cultivation. The distribution of different types of fatty acids was in agreement with the previous data reported on glucose and xylose [8], whereby oleate and palmitate where the most abundant at the end of batch cultivation. Notwithstanding the general agreement between both studies, in our study the PUFAs, mainly linoleate (C18:2) increased more during Nlim, while in the previous study

[8] palmitate (C16:0) increased more during lipid accumulation. It might possibly reflect the fact that different *R. toruloides* strains were used in these studies.

Final biomass titers were similar on xylose and glucose, respectively 18 g/L and 22 g/L (**S1 Fig**), but the highest biomass yield, 0.32 gDCW/g_substrate, was reached on xylose during Nlim phase (**S1 Table**). On xylose, 32% of substrate was excreted as byproducts xylitol and D-arabinitol during the exp growth phase (**Fig 1G**). For arabinitol, a stereoselective analysis was done using high-performance liquid chromatography (HPLC) separation with a chiral column (Chiralpak, Daicel Technologies, Japan), similarly as described in Lopes and Gaspar 2008 [22] (**S2 Fig**). Although Jagtap and Rao [23] already assumed the production of D-arabinitol, we were able to validate it. At low growth rates (during the Nlim phase), xylitol and D-arabinitol were not excreted but rather co-consumed. All byproducts were consumed at the end of the experiment at 168 h. On acetate, the amount of byproducts other than $CO_2$ increased during the Nlim phase to 31% (**Fig 1G**). These byproducts remain to be identified. On glucose, no byproduct other than $CO_2$ was detected. However, we were able to measure only 68.5% of carbon during the Nlim phase (**Fig 1G and S1 Table**). Likely, it was because *R. toruloides* strain CCT 7815 was making cell aggregates when grown in the chemically defined glucose-based medium. Mass balance calculation took into account glucose uptake, carbon dioxide production and biomass (in C-mol). As our biomass measurements were based on optical density, which relies on the assumption that cells are evenly distributed and of equal size [24,25], it may underestimate the actual cell concentration in liquid culture when aggregates are formed. To solve aggregation problem, we switched the nitrogen source from ammonium sulfate to urea in glucose condition. It helped to reduce aggregate formation but did not eliminate it. Based on the comparison of growth curves when using ammonium sulfate or urea, the results were highly similar (**S1A Fig**). Further analysis with glucose was carried out using urea as the nitrogen source. Therefore, results exclusively in the glucose condition (both exp and Nlim growth phases), including the proteomics and metabolic flux data presented in **Figs 1–5** and **S1–S13 and S1–S6 Tables and S1–S6 Datasets**, belong to experiments in which we used urea as a nitrogen source.

## Proteomics data shows a significant allocation into ribosomes

We also present a high quality dataset with absolute proteome abundances of *R. toruloides* measured at the late exp and mid-Nlim phases during growth on xylose, glucose and acetate. Proteins were measured and quantified with mass spectrometry-based TPA (total protein amount) quantification method [26], and we were able to determine the absolute abundances of 3160 proteins across 6 conditions (**S1 Dataset**). Principal component (PC) analysis showed coherency in our proteome data (**Fig 2A**). High similarity between acetate exp and Nlim data was detected, while showing significant differences with other studied conditions (separated on the PC1, describing 49% variation in the data). PC analysis has previously been done for *R. toruloides* strain IFO 0880 comparing gene expression during the exponential growth phase on rich medium containing sole carbon substrate, similarly as in our study [3]. PC1 using transcriptomics revealed distinct expression patterns on acetate-grown as compared to glucose- and xylose-grown cells, agreeing with the proteomics results obtained our study. The only noticeable difference was that PC2 in [3] separated exp phase from glucose to xylose. In our study, PC2, describing 34% of the variation in our data, separated mainly exp and Nlim conditions in the same way on glucose and xylose (**Fig 2A**).

Significant variation in proteome between the two growth phases, exp and Nlim, was also observed by differential expression analysis. We found 204 differentially expressed proteins in Nlim (lipid accumulation) versus exp growth phase on glucose, 37 on xylose and none on

acetate using a cut-off of |log2FC| > 1 and Benjamini-Hochberg corrected p-value < 0.05 (**S3 Fig** and **S2 Dataset**). Proteome profiles on xylose and glucose were more similar in comparison to the growth on acetate. Comparison of protein levels between carbon sources revealed the largest difference between xylose and acetate at exp growth phase, resulting in 159 differentially expressed proteins (**S3 Fig** and **S2 Dataset**). We then analyzed protein levels based on Gene Ontology (GO) group relations that represent different metabolic pathways present in *R. toruloides*. GO groups were obtained from the Uniprot database (*R. toruloides* NP11) and genome-scale model, rhto-GEM [15] (for a full list see **S2 Table**). GO relations from both sources were combined to provide possibly the most accurate information on different metabolic pathways present in *R. toruloides*.

We discovered that ribosomes formed the largest GO group of the proteome (data were represented per gram of total proteome, µg/g_protein) (**Fig 2B**). Interestingly, the ribosomal abundance in *R. toruloides* up to 46% of proteome was higher than observed previously in *S. cerevisiae* (around 37%) [27]. Expression levels in glycolytic pathways were largely unchanged during the lipid accumulation in xylose and acetate conditions, while upregulation was observed on glucose (**Fig 2B**). On glucose- and xylose-grown cells, proteome allocation to TCA cycle was considerably lower compared to glycolytic metabolic pathways (**Fig 2B**). On acetate, protein levels of the TCA cycle were almost 3-fold higher than on glucose and xylose (**Fig 2B**). Higher TCA cycle activity was expected as acetate assimilation directly produces TCA cycle-related metabolites. The electron transfer chain (ETC) was the only metabolic pathway, in which protein levels increased significantly during lipid production in all the studied carbon substrates (**Fig 2B**).

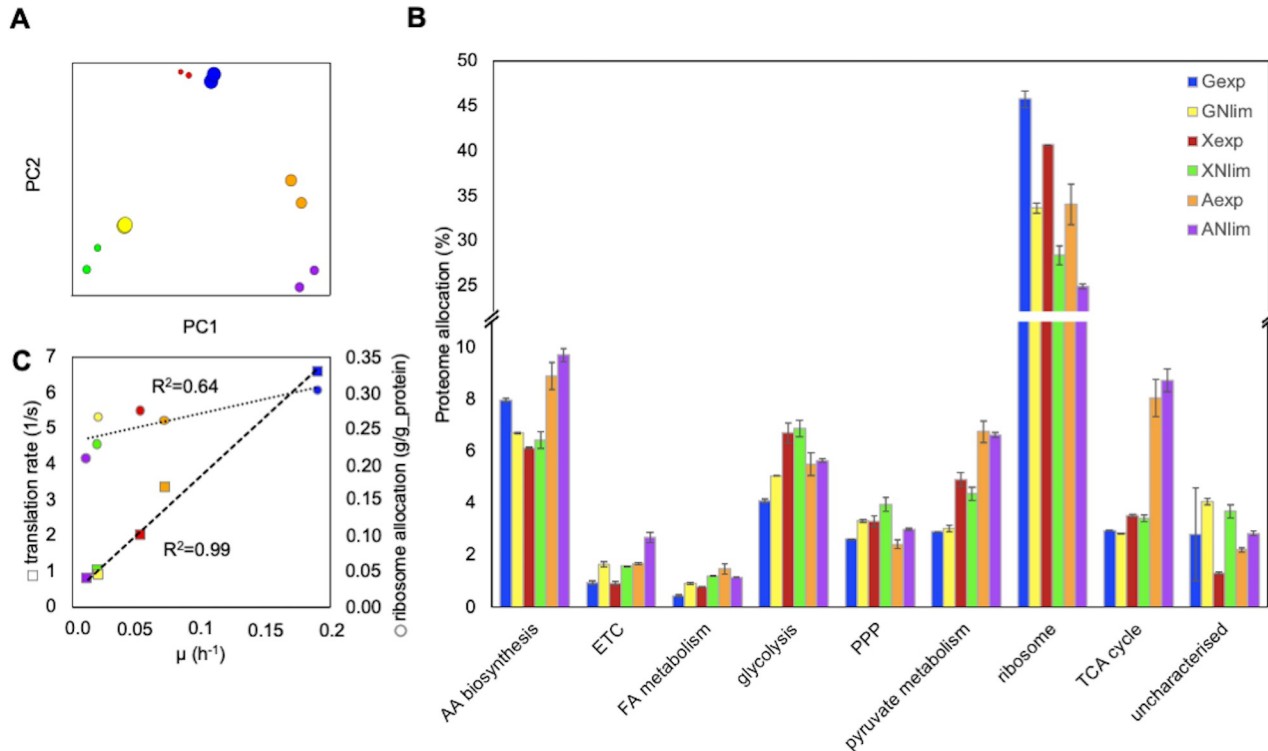

**Fig 2. Absolute proteomics data.** (A) Principal component analysis (µg/g_protein). (B) Proteome allocation (% of µg/g total protein) to metabolic pathways associated with amino acid (AA) biosynthesis, electron transport chain (ETC), fatty acid (FA) metabolism, glycolysis, pentose phosphate pathway (PPP), pyruvate metabolism, ribosome, tricarboxylic acid (TCA) cycle, and uncharacterized proteins. (C) Ribosomal translation rate (s$^{-1}$) and ribosome allocation (g/g_protein). Average of duplicate experiments with SD is illustrated. Proteins in each GO group are shown in **S2 Table**.

During lipid accumulation, the amounts of uncharacterised proteins, especially in xylose condition, increased (**Fig 2B**), indicating the importance of discovering unknown protein functions for future research. Interestingly, on glucose the least amount of proteins were allocated to FA metabolism, while the highest total lipid content was measured experimentally. The highest expression levels of proteins in the fatty acid metabolic pathways were detected on acetate (**Fig 2B**). It was mostly due to high expression levels of beta-oxidation proteins (**S1 Dataset**).

## Relation between ribosomal content, growth rate and translation

We used absolute quantification of the proteome and the ribosomal content to calculate the rate of protein synthesis per ribosome, also known as ribosome efficiency or protein translation rate (for instructions see **S3 Table**). The ribosome of *R. toruloides* strain NP11 was characterized by 178 structurally distinct proteins reported in Uniprot.org, from which 147 were identified in CCT 7815 strain and quantified (**S2 Table**). The calculated translation rates varied from 0.8 to 6.6 aa/s (**Fig 2C**), which was very similar as observed in *S. cerevisiae* (between 2.8 and 10 aa/s) [28]. Among the 6 conditions analyzed, we observed a linear correlation between the translation rate and specific growth rate μ ($R^2 = 0.99$, p-value $< 0.001$). The mass-wise ribosome content of proteome (g/g_protein) (**Fig 2C**) had no such distinct correlation with the μ ($R^2 = 0.68$, p-value $= 0.043$). Interestingly, the lowest ribosome content in proteome was detected during growth on acetate as compared to other substrates.

## Integrating fluxomic and proteomic analysis using an enzyme-constrained genome-scale model

Genome-scale models allow an *in silico* simulation of intracellular flux patterns in accordance with exchange fluxes obtained from cultivation experiments. To improve the predictive power and consider the capacity constraints imposed by enzymatic catalytic capacities and their protein levels, we developed an enzyme-constrained GEM using the GECKO Toolbox [17]. In lieu of a strain-specific model, we used the NP11-based GEM [15] to represent the CCT 7815 strain used in this study. The genome of its parental strain CCT 0783 possesses two versions of the same gene, one presenting >90% identity and the other version presenting >70% identity to the genome of haploid strain NP11 [29]. We integrated individual protein concentrations with their corresponding catalytic activities ($k_{cat}$) in the model to constrain individual metabolic fluxes. We created separate models for exp and Nlim phases on xylose (X), glucose (G) and acetate (A), respectively. Hence, 6 different versions of the proteome constrained model with modified biomass composition, fatty acid profiles and flux bounds from the experimental data were constructed. Proteome constraints included the concentrations of 773 different enzymes across all conditions (**S4A Fig and S4 Table**), which were applied to 1515 metabolic reactions (30% of all reactions) (**S5 Table**). The coverage of these constraints was greatly improved by manually assigning EC numbers to 461 *R. toruloides* enzymes (**S4 Table**), which enabled GECKO Toolbox to assign their $k_{cat}$ values. At first, BRENDA was queried for exact matching reaction, substrate and organism. But as kinetic parameter data for non-model organisms such as *R. toruloides* were not readily available, GECKO Toolbox step-wise relaxes the stringency when matching EC number, organism and substrate, to assign reasonable estimates of $k_{cat}$ values [30]. Mass-wise, the proteome constraints of the measured fraction of enzymes covered between 14% (Gexp) to 25% (ANlim) of the quantified proteome (**S4B Fig**). Aside from enzyme concentrations, proteome constraints contained 535 unique $k_{cat}$ values automatically queried from the BRENDA database (**S3 Dataset**). Models, data sets and scripts are hosted on a dedicated Github repository ecRhtoGEM (www.github.com/alinarekena/ecRhtoGEM).

Next we used Flux Balance Analysis [31] to simulate intracellular flux patterns and random sampling of the solution space [32] with 2000 sampling iterations to evaluate flux variability (**S4 Dataset**). To constrain the set of feasible solutions during sampling, we fixed the upper bound and lower bounds on the observed exchange fluxes, ATP hydrolysis (non-growth related maintenance) and protein pool exchange (see **Methods**). The average flux variability, estimated as a percentage of SD divided by the flux values, was 19% (median value of all conditions) (**S4 Dataset**).

From the simulated flux values, we calculated apparent enzyme catalytic activities ($k_{app}$) as ratio of model-predicted fluxes and measured protein concentration. $k_{app}$ represents the apparent *in vivo* enzyme turnover which drives the biological processes in the environment, in contrast to $k_{cat}$ representing maximum enzyme capacity. As we used model-predicted fluxes that were constrained by $k_{cat}$ values in the ec-model, the $k_{app}$ values that we obtained cannot be higher than the $k_{cat}$ value, which means that we cannot capture any potential *in vivo* enzyme activity enhancement effect. Regardless, in case of high $k_{app}$ values, high reaction rates are catalyzed by low protein concentration, and vice versa. This study is the first report on the *in silico* $k_{app}$ values in *R. toruloides*. Calculated $k_{app}$ values in all growth conditions are available in **S3 Dataset**. Vast majority of $k_{app}$ values were in the range from 0.1 to 100 ($s^{-1}$) (**S5 Fig**), which is in the range of "average enzyme" $k_{cat}$ of 10 $s^{-1}$ reported by Bar-Even et al. [33]. Some of the lowest $k_{app}$ values in acetate condition were associated with fatty acid degradation and beta oxidation metabolic pathways. We found that during the Nlim phase, when lipid accumulation occurs, the number of enzymes with relatively low $k_{app}$ values (0.1 to 1 $s^{-1}$) was increased (**S5 Fig**). This reflects the fact that absolute fluxes decreased more than protein concentrations during the Nlim phase in comparison to exp growth phase, suggesting that for many reactions downregulation of the enzyme did not affect its reaction rate directly.

## Growth on glucose

In our analysis of integrated flux and proteomics data, we focused on the major carbon fluxes and corresponding enzymes in the central carbon and lipid metabolism where acetyl-CoA, ATP and NADPH, the main precursors for lipid biosynthesis, are generated (**Fig 3**). For a better comparison, fluxes were normalized to the substrate uptake rate of the respective condition, providing percentage values of carbon distribution in the metabolic pathways (**S6 Fig**, for a full list see **S4 Dataset**). During the exponential growth phase on glucose, 72% of the carbon was directed via the PPP, while only 20% went through the Embden-Meyerhoff glycolytic pathway. When comparing fluxes at exp versus Nlim phase, we did not observe any significant changes in normalized fluxes through the oxPPP in glucose condition (**Fig 4A**), which was also the main source of NADPH regeneration (**S7 Fig**) (reaching 76% glucose-derived carbon). During the exp phase, the majority of NADPH was consumed by glutamate dehydrogenase (GDH) which converts ammonium and oxoglutarate (AKG) to glutamate, while during the Nlim phase majority of NADPH was consumed in lipid biosynthesis by FAS1-2 (**S7 Fig**).

The flux via phosphoketolase pathway, which converts D-xylulose 5-phosphate to glyceraldehyde 3-phosphate and acetyl-CoA, increased more than 4-fold from 14% to 60% during the transition from exp growth to Nlim phase, consistent with a significant upregulation of phosphoketolase (XPK) on proteome level (apval. 0.043, **S2 Dataset**). While it is not known, which route of XPK enzyme in combination with a phosphotransacetylase (PTA) or an acetate kinase (ACK) is used in *R. toruloides* strain CCT 7815, we compared the fluxes of both possible scenarios (**S4 and S5 Datasets**). As the results were highly similar, further flux analysis was carried out based on a metabolic route where PTA is active. XPK pathway was also the main

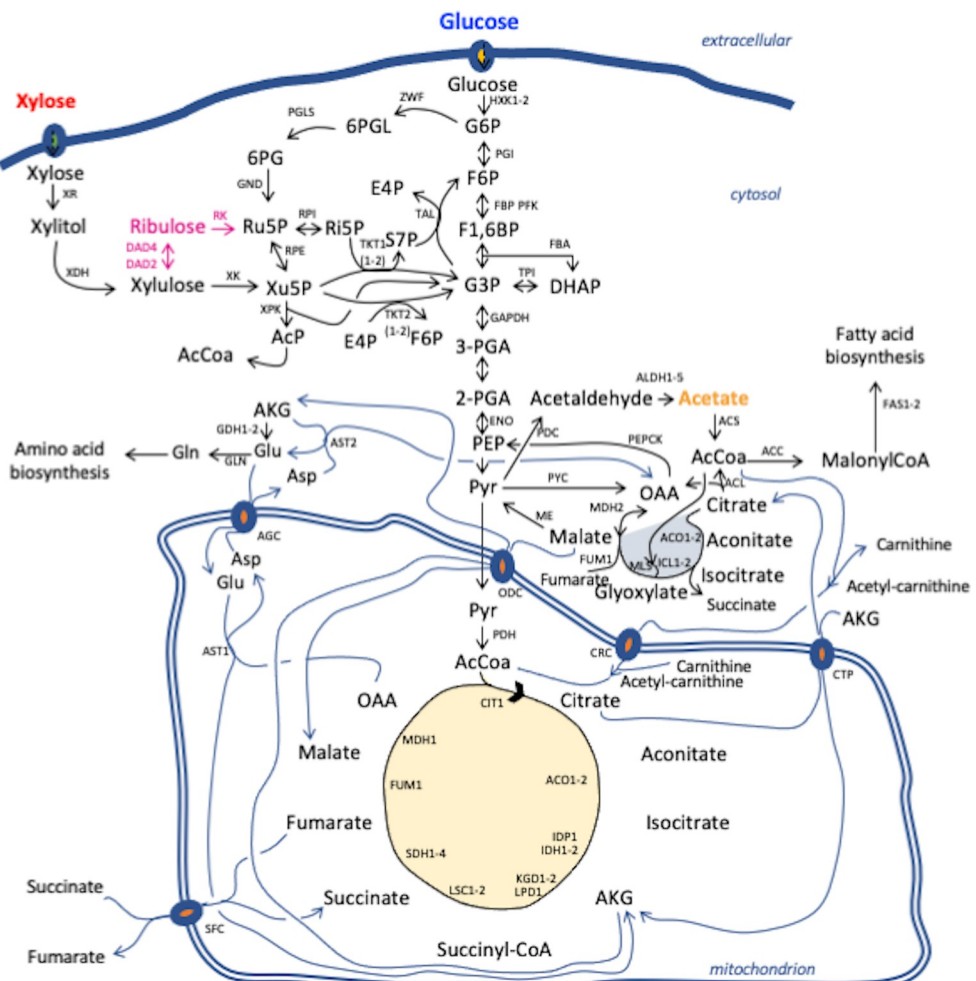

**Fig 3. Main metabolic pathways present in *R. toruloides*.** Blue arrows are used to denote mitochondrial carrier proteins and enzymes in shuttling pathways. Pink arrows are used to denote the alternative xylose assimilation pathway. Names, protein names and corresponding metabolic reaction IDs of genes are shown in **S2 Table**.

source of cytosolic acetyl-CoA during lipogenesis, which activity has never been reported on glucose, but is in line with previous findings in xylose condition [1,7]. The pyruvate decarboxylase and ACL, which exist as alternative pathways for producing cytosolic acetyl-CoA during lipid accumulation, were activated only when we blocked the XPK pathway (**S6 Dataset**). During the exp phase, cytosolic acetyl-CoA was not fully used for fatty acid biosynthesis, but 3% of carbon from glucose was transferred to TCA cycle via carnitine carrier (CRC) via acetylation reaction and in exchange of carnitine. The transfer of acetyl-CoA to mitochondria likely reflects that there was sufficient availability of cytoplasmic acetyl-CoA during the exp phase on glucose. At Nlim phase, majority of cytoplasmic acetyl-CoA was consumed by acetyl-CoA carboxylase (ACC), the first step in lipid biosynthesis, as more than 5-fold increase between 9% to 58% of carbon was observed via ACC during the transition from exp phase.

The main flux from the pyruvate branching point was channeled to the TCA cycle via pyruvate dehydrogenase (PDH), reaching 69% of carbon from glucose during exp phase. During exp phase, 37–46% of glucose-derived carbon was channeled from aconitase (ACO1-2) to fumarase (FUM1), while the flux through malate dehydrogenase (MDH1) appeared to

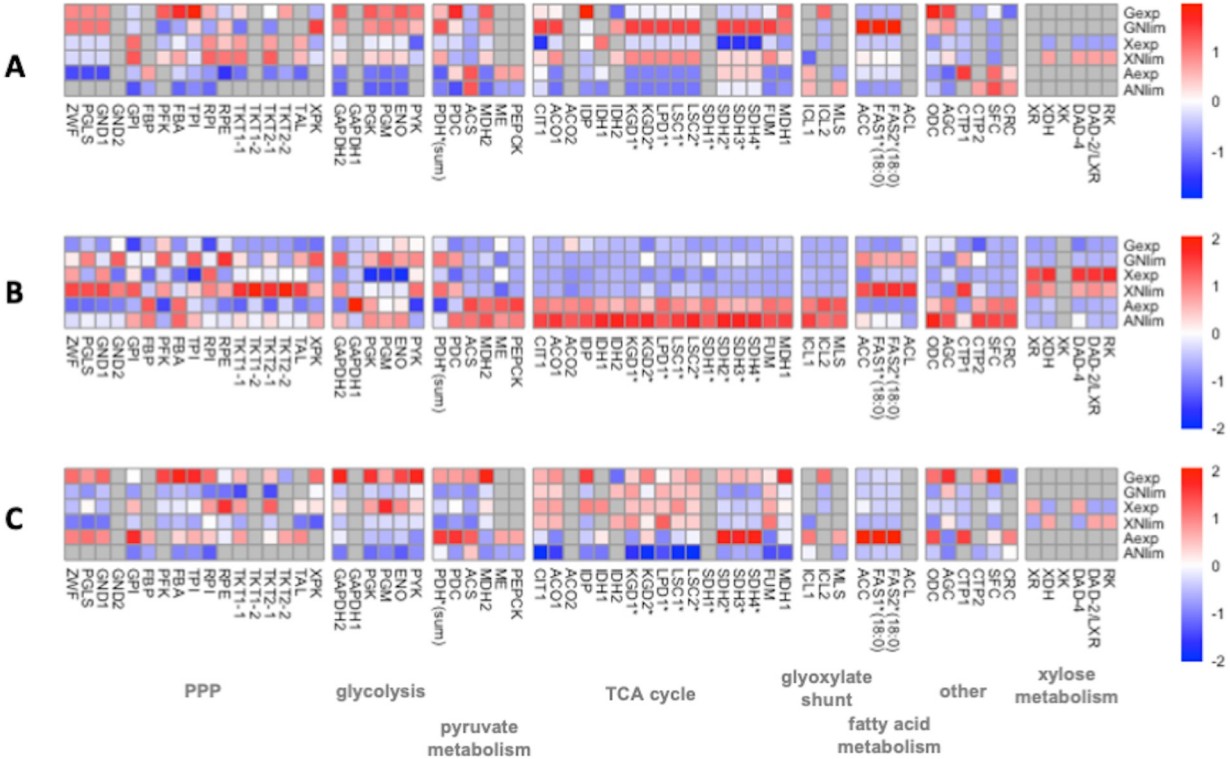

**Fig 4. Heatmap of Z-scores for enzymes in central carbon metabolism.** (A) Fluxes (mmol/gDCW/h) calculated as median from 2000 iterations of random sampling of the solution space [Bordel et al. 2010] [32], normalized by substrate uptake rate. (B) Proteomics data (μg/g of total protein) calculated as the average of duplicate experiments. (C) Apparent catalytic activities, $k_{apps}$ ($s^{-1}$), calculated as fluxes divided by protein concentrations. Gray color is used to denote missing values. Asterisks (*) are used to denote isoenzymes. PPP: pentose phosphate pathway. Names, protein names and corresponding metabolic reaction IDs of genes are shown in **S2 Table**.

correspond to 218% of glucose-derived carbon, indicating that an internal cycling of carbon was taking place. Similar recycling has been noticed also before in *R. toruloides* genome-scale models [1,7,11], however we assume this cycling to be artificial caused by a high demand of mitochondrial NADH. During the Nlim phase, ATP yield increased (**S8B Fig**), consistent with a significant upregulation of ETC on proteome level (**Fig 2B**). Since the TCA cycle and the respiratory chain are metabolically connected, cells require mitochondrial NADH to drive respiration. Ultimately, the yield of NADH during the Nlim phase slightly increased (**S9B Fig**). Fluxes through the TCA cycle significantly increased (**Fig 4A**), reaching 83% of carbon from glucose (**S6 Fig**), while the biomass yield decreased (**S1 Table**).

With respect to intracellular protein levels, the concentration of ACL was also high (1010 ±75 μg/g_protein) at exp phase and was 2.6-fold upregulated (apval. 0.039) during the Nlim phase (**Fig 4B** and **S2 Dataset**), which was consistent with results from previous proteomics studies suggesting PDH-CIT1-ACL path for producing cytosolic acetyl-CoA [8,10], but not supported by the model flux results. The concentrations of PDH (2304±11 μg/g_protein) and CIT1 (2573±6 μg/g_protein) (**S10 Fig**) at exp phase were higher than on average for the TCA cycle enzymes (coinciding with high $k_{cat}$ values of 486 $s^{-1}$ and 540 $s^{-1}$ for PDH and CIT1, **S3 Dataset**), which was also consistent with their role in the PDH-CIT1-ACL route, suggested by previous omics studies. However, the expression levels of mitochondrial membrane carrier proteins at both growth phases were low. At relatively low protein levels, high flux through these transporter proteins resulted in higher $k_{app}$ values (**S11 Fig**).

## Growth on xylose

Next, we explored *R. toruloides* metabolism during growth on xylose. Xylose is metabolized by xylose reductase (XR, NADPH-dependent), which reduces xylose to xylitol, xylitol dehydrogenase (XDH) and xylulokinase (XK), and further assimilated into central carbon metabolism via transketolase (TKT1-2) or XPK pathway. The expression of XK was not detected on proteome level in any of the conditions studied, suggesting an alternative pathway to the known fungal xylose assimilation pathway (**Fig 3**). The experimental detection of D-arabinitol isoform suggested the conversion of D-xylulose to D-arabinitol. This mechanism was supported by the presence of two genes in the *R. toruloides* genome encoding D-arabinitol dehydrogenase, RHTO_07844 and RHTO_07702. Only protein RHTO_07844 was detected in our proteomics analysis (1913 μg/g_protein) (**S10 Fig**), suggesting its role as D-arabinitol 4-dehydrogenase (DAD-4), converting D-xylulose to D-arabinitol. Arabinitol dehydrogenase could also convert arabinitol to ribulose (D-arabinitol 2-dehydrogenase) [34]. L-xylulose reductase (LXR) of fungal *A. monospora* has been reported to reversibly convert D-ribulose to D-arabinitol [35]. In support of this mechanism, protein levels of L-xylulose reductase (EC 1.1.1.10, RHTO_00373) were 10-fold upregulated during growth on xylose versus other substrates. Therefore, RHTO_00373 was selected as D-arabinitol 2-dehydrogenase (DAD-2) (converting D-arabinitol to D-ribulose). Arabinitol dehydrogenase is known to use NADH as cofactor [34]. LXR is mostly known for NADP(+)/NADPH specificity [35]. D-ribulose can enter the non-oxidative part of PPP via phosphorylation by D-ribulokinase (RK). An equivalent pathway was recently reported by Kim et al. 2021 [9]. One gene in *R. toruloides* IFO 0880 GEM (version 4.0) was annotated as D-ribulokinase (ID 14368) and we used it to identify potential RK in NP11 strain, which is more similar to the strain CCT 7815 used in this study [29]. Gene RHTO_00950 was identified as an ortholog of protein ID 14368 by a BLAST search, which found a match with 98.5% identity. Interestingly, in strain IFO 0880 orthologs of both genes RHTO_07844 and RHTO_07702 were identified as DAD-2 and DAD-4, respectively, and both were using NAD/NADH as the cofactor [9]. While it is not known, which cofactor of DAD-2/LXR enzyme is operational in *R. toruloides* strain CCT 7815, we analyzed the fluxes of both possible scenarios. Both simulation results were highly similar, with a difference in where NADPH was regenerated. The alternative pathway through DAD was preferred even when XK was not constrained with proteome.

In a scenario when DAD-2/LXR was NADP-dependent, during both exp and Nlim growth phases between 46–49% of carbon derived from xylose was directed via glucose 6-phosphate isomerase (GPI) in a reverse direction to the glycolytic flux. In a combination with that, 42% of carbon was directed via the oxPPP and returned to the Ru5P branching point, indicating that a loop associated with NADPH recycling is taking place. Alternatively, up to 88% of xylose-derived carbon was directed via oxPPP (**S7 Dataset**). In the first scenario, ZWF and GND provided more NADPH than LXR/DAD-2 during the exp phase (**S12 Fig**). During the Nlim phase, when the yield of NADPH slightly increased (**S13 Fig**), the flux of ZWF and GND remained unchanged, while the flux of LXR/DAD-2 increased (**Fig 4A**). XR consumed at least 2-fold more NADPH than any other NADP(+)-dependent enzyme during both growth phases. However during Nlim, more NADPH consumed by FAS1-2 was spent on lipid biosynthesis (**S12 Fig**).

From the proteomics analysis, the concentrations of enzymes involved in the xylose pathway were 1.1 to 1.6-fold downregulated during Nlim phase versus exp phase (**Fig 4B**), consistent with the decrease in xylose uptake rate (**S1 Table**). Lower concentration of RK (644±8 μg/g_protein) compared to other enzymes involved in xylose assimilation was measured (**S10 Fig**), suggesting enzyme limitation in the XK bypass pathway. At relatively low protein levels, high flux through RK resulted in relatively higher $k_{app}$ values (**S11 Fig**).

Aside from enzymes directly involved in xylose assimilation, the intracellular flux patterns on xylose were the closest to growth on glucose, in comparison to growth on acetate. The flux of XPK was upregulated 1.7-fold between 13% to 22% during the Nlim phase (**Fig 4A**), which was also the main source of acetyl-CoA during lipogenesis. At the Nlim phase, the yields of ATP and NADH significantly increased (**S8 and S9 Figs**). The additional mitochondrial NADH during the Nlim phase was provided via internal cycling of MDH1 (**S14 Fig**).

## Growth on acetate

Lastly, we explored *R. toruloides* metabolism during growth on acetate. Acetate can cross the plasma membrane to enter the cells via simple or facilitated diffusion, but at pH below neutral (< pH6) the diffusion of the undissociated form of the acid induces the stress response or causes negative effect on metabolic activity [36]. In *R. toruloides*, two permeases have been found upregulated during growth on acetate-based rich medium in comparison to glucose-based rich medium [3], suggesting that facilitated diffusion is taking place. Once inside the cells, acetate is assimilated via ACS that directly provides acetyl-CoA (**Fig 3**), one of the main precursors for lipid biosynthesis. From acetyl-CoA branching point, the flux is channeled into the central metabolic pathways via isocitrate lyase (ICL1-2) and malate synthase (MLS), which are predicted to be located in cytosol, but no experimental evidence is available. Metabolic model predicted that at acetyl-CoA branching point, 18% of carbon from acetate during exp growth phase was directed to lipid biosynthesis via ACC, while the majority of carbon (51%) entered glyoxylate shunt. In addition, a significant amount of carbon from acetyl-CoA (29%) was channeled via CRC carrier, which was predicted to have a minor activity on glucose condition. The CRC route was preferred over the PDH pathway towards mitochondrial acetyl-CoA (MLS-ME-PDH), which channeled only 18% of carbon from acetate at exp phase. Metabolic model predicted 5% of carbon from acetate excreted as succinate from the glyoxylate shunt, in addition to 2% of carbon excreted as citrate, which was confirmed by HPLC. During the Nlim phase, the main fluxes demonstrated different regulation (**Fig 4A**). The increase in flux via CRC (1.4-fold) reflects that more carbon entered the TCA cycle during the Nlim phase. Interestingly, the flux of ACC was downregulated 3.1-fold at Nlim compared to the exp growth phase. Using the rate of lipid production, which decreased during Nlim phase, it can be explained that the lipid production in absolute amounts was higher during the exp phase to sustain the growth together with moderate lipid production (**S1 Table**).

On acetate, fluxes of the TCA cycle were the lowest, while measured protein levels were the highest among all conditions analyzed (**Fig 4A and 4B**). At Nlim phase, 28% of carbon from acetate was predicted to be excreted as OAA, while the biomass yield decreased (**S1 Table**). Flux levels of the TCA cycle indicated that an internal cycling of carbon similar as in other conditions was taking place (**S6 Fig**). It involved different transporter proteins,—the citrate-oxoglutarate (CTP) and succinate-fumarate (SFC) transport -, which allow channeling of the flux from the TCA cycle to glyoxylate shunt (**Fig 3**). During the Nlim phase, ATP turnover, produced entirely via ETC (**S15 Fig**), and NADH turnover, produced almost entirely via the TCA cycle (**S16 Fig**), both decreased (**S8 and S9 Figs**), unlike observed in glucose or xylose conditions, where ca 80% of the ATP originated from ETC, while the rest came mainly from glycolysis.

Aside from enzymes directly involved in the TCA cycle, cytosolic ME was the sole supplier of NADPH during the Nlim phase only in acetate condition (**S17 Fig**). This is supported also by the measured protein levels of ME, which were significantly higher under acetate conditions, although the absolute levels of ME were relatively low under all studied conditions (189 ±3 μg/g_protein) (**Figs 4B and S6**). Only during the growth on acetate the NADPH yield

decreased during the Nlim phase (**S13 Fig**). During the Nlim phase, more NADPH was consumed by FAS and spent on lipid biosynthesis (**S17 Fig**).

We also observed few significant changes in fluxes of enzymes involved in gluconeogenesis, which is an important pathway during growth on acetate to provide xylose phosphate-based precursors for ribonucleotide synthesis. The normalized flux towards gluconeogenesis, channeled via MDH2, carried 11% of carbon exp growth phase. It may reflect the fact that PEPCK, the first enzyme in the gluconeogenesis pathway, consumed ATP, but we found that PEPCK was consuming only 2.4% of ATP during exp phase (**S15 Fig**). The concentration of PEPCK (627 µg/g_protein) (**S10 Fig**) and its $k_{cat}$ value (38 s$^{-1}$) (**S2 Dataset**) were low, suggesting that PEPCK could have been a rate-limiting step of gluconeogenesis during the exp phase.

From proteomics analysis, the concentration of enzymes involved in fatty acid beta oxidation (RHTO_04957, RHTO_00300, RHTO_02848, RHTO_07118, RHTO_00476) at higher concentrations (from 163±9 to 531±7 µg/g_protein) as compared to cells grown on other substrates at exp phase (**S2 Dataset**), suggesting this pathway might be more active in *R. toruloides* during growth on acetate.

## Intracellular flux patterns point to metabolic trade-offs associated with lipid production

Multi-layer data provided in this study allows us to analyze metabolic trade-offs and compare the resource allocation between different metabolic pathways present in *R. toruloides*. From the metabolic modeling results, we analyzed the NADPH allocation between nitrogen assimilation and fatty acid biosynthetic pathways in glucose- and xylose-grown cells (**Fig 5**). When the yeast had an abundant source of nitrogen, either by conversion of urea to ammonia (glucose condition) or by growth on ammonia itself, NADP-dependent glutamate dehydrogenase (GDH1) converted ammonia with the TCA cycle intermediate AKG into glutamate, which was then used for the amino acid biosynthesis. Thus, during the exp phase on glucose-grown cells 46% of NADPH turnover was channeled via GDH1, while 13% was consumed via FAS1-2. On the contrary, during the Nlim phase 12% of NADPH was channeled via GDH1, while 75% was consumed via FAS1-2. This was supported also by the measured protein levels of GDH1, which were significantly higher during exp growth phase (4294±183 µg/g_protein), as compared to Nlim phase (2948±135 µg/g_protein) (**S2 Dataset**). In this metabolic trade-off, less (almost 4-fold) cytosolic NADPH was consumed by GDH1 during the Nlim phase (**S7 Fig**) when the protein content reduced 2.8-fold (**S1 Table**). And vice versa, more NADPH (almost 6-fold) was consumed by FAS1-2 when total lipids increased 8.5-fold.

In acetate condition, the metabolism during lipogenesis at Nlim phase might be influenced by the beta-oxidation, a metabolic process of lipid degradation that can return carbon back into central metabolism, which was detected on proteome level in acetate condition. But the activity of this pathway could not be simulated with our current model.

## Discussion

In this study, we presented detailed analysis of physiological characterization of *R. toruloides* CCT 7815 during growth on glucose, xylose or acetate as a sole carbon source. It was an important part of the study as the collected data together with the quantitative proteomics analysis was used to constrain the newly developed enzyme-constrained metabolic models. Cultivation experiments were carried out at a C/N ratio, which allowed nutrient excess conditions during the first part of the batch cultivation and resulted in nitrogen limitation during the second part of the experiment, a growth phase known to induce lipid accumulation [12]. Enabled by bioreactors equipped with online monitoring sensors, we were able to accurately

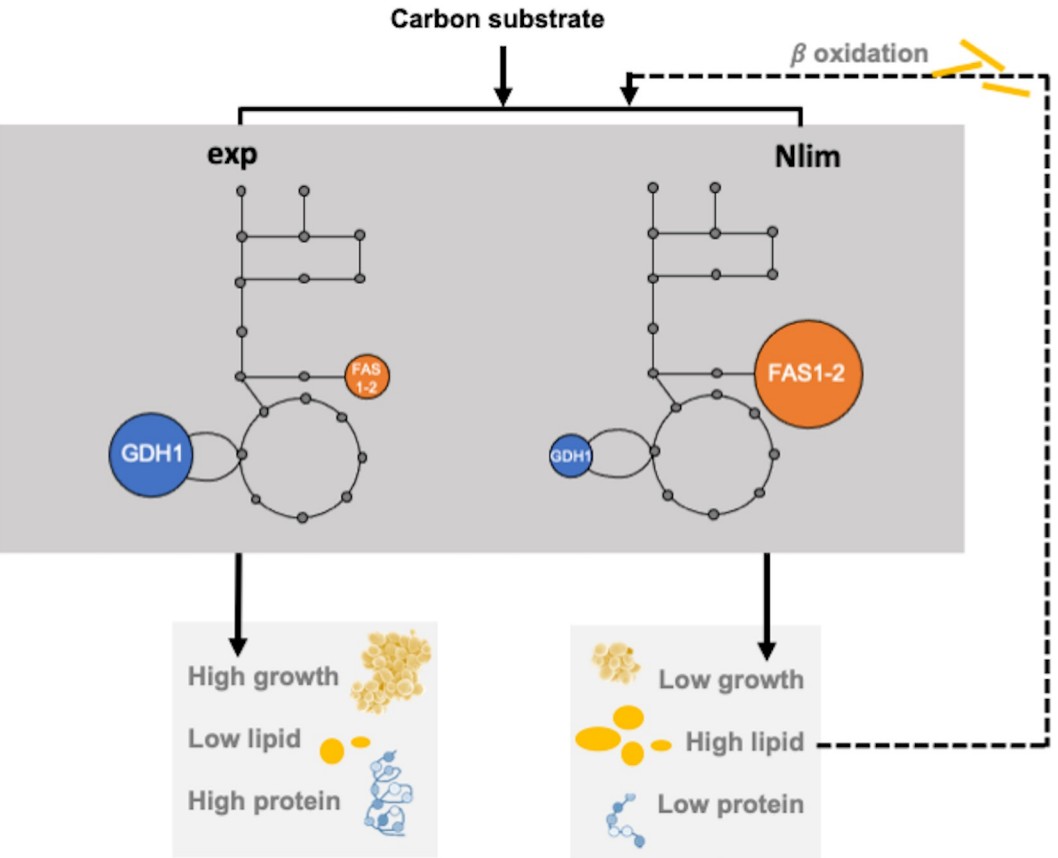

**Fig 5. The trade-off between NADPH expenditure in *R. toruloides*.** Blue circles represent nitrogen assimilation and orange circles represent lipid biosynthetic pathways of *R. toruloides* cultivated on glucose during exponential growth (exp) and nitrogen limitation phase (Nlim). Circle size (blue and orange) represents the % of NADPH turnover. GDH1, glutamate dehydrogenase (NADP+); FAS1-2, fatty acid synthase (α and β).

measure two distinct growth phases during the batch culture (**S1 Fig**), characterized by different lipid content in biomass and other physiological parameters (**S1 Table**), which were later implemented into metabolic models.

Experimental results showed a slightly higher final biomass yield (0.32 gDCW/g_substrate) during growth on xylose as compared to Tiukova et al. 2019, [8] (0.28 gDCW/g_substrate), where significantly lower starting sugar concentration (40 g/L) and a different *R. toruloides* strain were used. Differences between strains were demonstrated also in their lipid composition (**Fig 1E**). Lower final biomass yield during growth on glucose (0.24 g/g) as compared to xylose condition (0.32 g/g) was difficult to explain by any other reason than the formation of cell aggregates on glucose, as no byproducts in this condition were detected. It has been reported earlier, and for future research aggregate formation in *R. toruloides* could be mitigated with the increased salt concentrations [37]. The possible explanation could be the production of exopolysaccharides that has been found in other *Rhodotorula* species [4]. However, the final lipid content in biomass during growth on glucose (48%) was the same as what has been reported in Tiukova et al. 2019, [8].

Absolute proteome quantification helped to improve the understanding of metabolism during lipid accumulation and on various substrates. Interestingly, we found that the proteome was largely unchanged during the Nlim versus exp phase on acetate, while up to 204 differently

regulated proteins were found on other tested substrates. Low proteome allocation to TCA cycle as compared to glycolytic metabolic pathways in the presence of glucose and xylose (**Fig 2B**) could possibly signal about citrate accumulation and subsequent transportation event from mitochondria to cytoplasm developed for lipid production.

Metabolic modeling confirmed previous modeling findings obtained on PPP pathway and ME as the main suppliers of NADPH, and XPK pathway as the primary source of acetyl-CoA during lipid biosynthesis in *R. toruloides* [1,7]. We found that these are carbon source dependent and to close proximity of the carbon substrate uptake. We also found that the pathways involved in synthesis of lipid precursors were not changed during the Nlim versus exp growth phase. To enable this analysis, we modified models' biomass reactions to reflect the measured lipid and protein content, based on precise physiology data of two distinct growth phases. For further work, it would be interesting to understand if the predicted fluxes via oxPPP while cells were growing on glucose and xylose are thermodynamically feasible.

Metabolic modeling also helped to explain *R. toruloides* physiological characteristics and byproduct excretion. Changes in biomass yield during growth on glucose were associated with increased fluxes through phosphoketolase (XPK) pathway and the TCA cycle, resulting in more carbon entering the TCA cycle during lipid accumulation. Predicted acetate kinase activity demonstrated certain robustness towards the preference for the XPK pathway (**S5 Dataset**). XPK pathway is considered an attractive option for generating cytosolic acetyl-CoA because it is more energy and carbon efficient. It circumvents one molecule of $CO_2$ lost per pyruvate and two equivalents of ATP consumed compared to the *PDH bypass* route (PDC-ALDH and ACS). In earlier studies using heterologous expression of XPK pathway in *S. cerevisiae* [38,39] it was found that the engineered strain had an increased flux towards TCA cycle and lower flux from the pyruvate branching point towards acetate formation. In a study by Bergman et al. 2019, [40], it was found that the activity of XPK pathway increases acetate flux and ATP requirement in *S. cerevisiae*, leading to an increased production of $CO_2$ and negative growth effects. Apart from the fact that there is no acetate excretion in *R. toruloides*, it would be interesting to further understand if use of XPK pathway in lipogenesis may also explain the carbon losses on glucose as observed in our study.

During growth on acetate, it was suggested that the byproduct formation was associated with energy metabolism, as the predicted excreted metabolites were TCA cycle-related. Using the metabolic models, we predicted a higher ATP turnover in acetate as compared to xylose condition (**S8 Fig**), while the specific growth rates were comparable. Our simulation results could not explain why Nlim phase, when more carbon entered the TCA cycle, was associated with the increase in internal cycling for NADH transport, also known as malate-aspartate NADH shuttle [41]. Stoichiometry in the oxidative phosphorylation pathway in *R. toruloides* in rhto-GEM has been adopted from experimental data in *S. cerevisiae* and was not changed in the present study. Moreover, rhto-GEM and ecRhtoGEM are based on *R. toruloides* exhibiting not only proton-pumping complex I (t_0001), but also external NADH dehydrogenase (r_0770) that do not pump protons. However, presence of complex I creates an electron competition process, which might have consequences on the yield of oxidative phosphorylation, as experimentally demonstrated in another Crabtree negative yeast *Candida utilis* [42]. As no experimental phosphate/oxygen (P/O) ratio has been yet reported in *R. toruloides*, it might be that the mitochondrial shuttling loop observed in our simulations in reality could correspond to a higher ATP requirement.

Internal carbon cycling was suggested not only in acetate, but also in glucose and xylose conditions. This mitochondrial shuttling loop was also noticeable from the results of earlier modeling studies [1,7,11]. In Dihn et al. 2019, [11], it was called "NADH shuttle", but it is yet to be confirmed if the shuttle could be an artificial loop to feed NADH to mitochondria.

Despite progress in understanding the physiology and genetics of *R. toruloides*, very little is known about mitochondrial carrier (MC) proteins in this strain. From experimental studies in *S. cerevisiae*, oxodicarboxylate (ODC) and aspartate/glutamate (AGC) carriers are important to export AKG (in exchange of malate) for nitrogen assimilation and also for the malate-aspartate NADH shuttle [43]. In support of this mechanism, our simulations on glucose-grown cells showed that the NADH produced via glycolysis was transferred to mitochondria for electron transport using enzymes MDH2 and ODC. Alternatively, it might point to an artifact of a different P/O ratio.

Metabolic models on xylose were greatly improved by the detection of chirality of D-arabinitol. We presented an alternative xylose assimilation pathway, which was favored in our model simulations over the known xylose pathway in fungi that involved D-xylulose 5-phosphate. Our results were consistent with recent reports in strain IFO 0880 [9], but we also detected some differences in the pathway, which were related to the fact that we used a different *R. toruloides strain*, CCT 7815. To explain byproduct formation upstream glycolysis while cells were growing on xylose, we presented several ideas associated with the energy and lipid metabolism. In comparison to our previous work [1], the flux of PPP could be compensated by the amount of carbon channeled via the alternative xylose pathway.

The fact that no carbon was directed via PDH-CIT-ACL pathway might also point to lack of alternative routes of NADPH regeneration in *R. toruloides*. In our results, the activity of oxPPP was coupled to an active XPK pathway supplying the vast majority of cytosolic acetyl-CoA during the Nlim phase. Our proteomics data showed a significant increase in the amount of uncharacterized proteins during lipid accumulation, especially in xylose condition (**Fig 2B**). Hypothetically, $CO_2$-decoupled NADPH regeneration would reduce the fluxes through oxPPP and XPK pathways (but not eliminate them), the flux of glycolysis would remain the same as shown in this study, but more carbon would be channeled via IDP and ACL. It has been demonstrated that GAPDH contributes to NADPH supply in filamentous fungi *Mortierella alpina* [44]. $CO_2$-decoupled NADPH synthesis has been engineered in *S. cerevisiae*, demonstrating significant phenotypic changes [45].

Modeling results revealed metabolic trade-offs associated with NADPH allocation between nitrogen assimilation and lipid biosynthetic pathways. In *S. cerevisiae* and *E. coli*, a clear specific growth rate dependence of ribosomal proteins has been demonstrated [27,46,47]. In the present study, we demonstrated a similar significant correlation for specific growth rate dependent ribosomal content (**Fig 2C**). Moreover, we were able to demonstrate a trade-off in NADPH demand. Although NADPH regeneration was dependent on the carbon source, NADPH demand was shifting from protein production at higher growth rates to lipid biosynthesis at lower growth rates in Nlim phase (**Fig 5**). Enzyme-constrained metabolic models developed in this study used not fully matched $k_{cat}$ values that can notably increase the prediction errors [48]. Characterizing enzymatic properties using physically based models requires enormous experimental work, therefore accurate computational approaches are needed to address this gap. Deep learning algorithms have demonstrated outstanding success in predicting protein structures based on their sequence information [49–51], and the method has also been applied in predicting enzyme $k_{cat}$ values for yeast *S. cerevisiae* [52].

## Conclusion

In this study, enzyme-constrained genome-scale metabolic models were generated for *R. toruloides*, where metabolic modeling together with proteome data gave a detailed interpretation of how flux patterns are changing in *R. toruloides* on different substrates during the exponential growth and in lipid accumulation. The results were consistent with previous knowledge on

the main pathways involved in lipid biosynthesis in *R. toruloides*, revealed by genome-scale modeling and multi-omics analyses. While detailed analysis of simulated intracellular flux patterns allowed us to explain some physiological parameters during growth on glucose, many observations require further validation. This work contributes to improving the knowledge about *R. toruloides* metabolism.

## Materials and methods

### Strain, media and growth conditions

*R. toruloides* CCT 7815 (Coleção de Culturas Tropicais, Fundação André Tosello, Campinas, Brazil) from a previous study [21] was used in the cultivation experiments. The same study identified increased lipid production, induction of hydrolysate-tolerance and lipid accumulation genes without physiological changes regarding growth and substrate consumption in *R. toruloides* strain CCT 7815 after a short-term adaptation in sugarcane bagasse hemicellulosic hydrolyzate. Seed cultures were grown on chemically defined medium according to Verduyn (3.0 g/L KH2PO4, 0.5 g/L MgSO4·7H2O, 15 mg/L EDTA, 4.5 mg/L $ZnSO_4$·7H2O, 0.3 mg/L $CoCl_2$·6H2O, 1 mg/L $MnCl_2$·4H2O, 0.3 mg/L $CuSO_4$·5H2O, 4.5 mg/L $CaCl_2$·2H2O, 3 mg/L $FeSO_4$·7H2O, 0.4 mg/L $Na_2MoO_4$·2H2O, 1 mg/L $H_3BO_3$, 0.1 mg/L KI, 0.05 mg/L biotin, 1 mg/L calcium pantothenate, 1 mg/L nicotinic acid, 25 mg/L inositol, 1 mg/L thiamine HCl, 1 mg/L pyridoxine HCl, 0.2 mg/L *para*-aminobenzoic acid [53]) supplemented with a sole carbon source of 18.2 g/L glucose, 20 g/L xylose or 20.0 g/L acetic acid and 5 g/L $(NH_4)_2SO_4$ in duplicate shake flasks at 200 rpm and 30˚C for 24 h. The carbon/nitrogen (C/N) molar ratio of the medium in seed cultures was 8.8. To obtain seed cultures, cells were pre-cultured in YPD media, and pelleted and washed twice with 0.9% (m/v) NaCl solution before inoculation. Seed cultures were used to inoculate 900 mL of chemically defined medium supplemented with either 63.6 g/L glucose and 0.9 g/L urea, or 70 g/L xylose and 2 g/L $(NH_4)_2SO_4$ or 20.0 g/L acetic acid and 0.6 g/L $(NH_4)_2SO_4$, and 0.1 mL/L antifoam 204 (Sigma-Aldrich, St. Louis, MO, United States) in duplicate bioreactors with a starting OD600 of 0.4 at 400–600 rpm, 30˚C, pH 6.0. At the start of cultivation, the (C/N) molar ratio of the media in bioreactors was set to 69 (glucose/urea) and 80 (xylose- or acetate/$(NH_4)_2SO_4$). Note, that xylose condition was carried out in Pinheiro et al. 2020, [1].

Cells were grown in 1-L bioreactors (Applikon Biotechnology, Delft, the Netherlands) in a batch cultivation regime. pH was controlled by the addition of 2 mol/L KOH. Dissolved oxygen was maintained not lower than 25% at 1-vvm airflow by regulating the stirring speed. CO2 and O2 outflow gas composition were measured using an online gas analyzer (BlueSens gas sensor GmbH, Herten, Germany). Cell turbidity was monitored on-line using Bug Lab BE3000 Biomass Monitor (Bug Lab, Concord, CA, United States) at 1300 nm and off-line using UV/Vis spectrophotometer at 600 nm (U-1800, Hitachi High-Tech Science, Tokyo, Japan). Data collection and processing was performed with BioXpert V2 software **v2.95** (Applikon Biotechnology, Delft, the Netherlands).

For dry cell weight measurement, samples were collected every 6 hours during the exponential growth phase and every 24 hours during the nitrogen limitation phase. For other analyses, samples were collected every 3 hours during the exponential growth phase and every 24 or 48 hours during the nitrogen limitation phase.

For extracellular metabolites, lipidomics and protein content analyses, samples were taken from bioreactors to 2-mL tubes, centrifuged for 30 s at 4˚C and 18000×g. The supernatant was stored at -20˚C for extracellular metabolite analyses. Cell pellets were snap-frozen in liquid nitrogen and stored at -80˚C for further analyses.

## Analytical methods

For dry cell weight (DCW) measurement, culture samples were taken from bioreactors to 2-mL tubes, passed through a 0.3 μm filter, dried and analyzed by gravimetric method. Biomass optical density data were calibrated by gravimetric cell mass measurements. For extracellular metabolites measurements, high-performance liquid chromatography (HPLC) separations were performed with Shimadzu instruments (LC-2030C Plus, Shimadzu, Kyoto, Japan) equipped with a refractive index detector (RID-20A, Shimadzu, Kyoto, Japan). Glucose, xylose, organic acids and glycerol concentrations were measured using a Rezex ROA Organic Acid column (Phenomenex, Torrance, United States). Separations were performed at 45˚C and the mobile phase for isocratic elution was 5 mmol/L $H_2SO_4$. The flow rate was 0.6 mL/min. Stereoselective HPLC analysis of arabinitol isomers was done using a Chiralpak column (Daicel Technologies, Japan) and the mobile phase for isocratic elution was a mixture of hexane and ethanol (70:30, v/v) at 20˚C; the flow was 0.3 mL/min. Chiralpak column of arabinitol standards gave different retention times for each enantiomer (D and L) (S18 Fig). Yields and specific consumption and production rates represent exp and Nlim phases separately, not cumulatively.

For intracellular total protein quantification, cell pellets were thawed on ice and resuspended in 0.9% (m/v) NaCl solution to a concentration of 1 g/L. Then 600 μg of biomass was mixed with a commercially available protein extraction solution (Y-PER, Thermo Fisher) in a 2-mL tube and incubated at 30˚C for 45 minutes. After incubation, samples were transferred to screw cap 2-mL tubes with glass beads. Cell lysis was performed using a FastPrep-24 device for 4 cycles (4 m/s for 20 s) with a 5 min interval after each cycle. After cell lysis, the tubes were centrifuged at 14800 rpm for 10 min at 4˚C. Supernatant was collected to a new 2-mL tube and the leftover biomass sample was subjected to a repeated extraction cycle (without 45 min incubation interval) until no proteome was detected in supernatant. Before quantification, all fractions of supernatant were combined. Proteome was quantified using a commercially available colorimetric assay kit (Micro BCA Protein Assay Kit, Thermo Fisher Scientific, Waltham, MA, United States). Protein concentration was determined using the calibration curve of bovine serum albumin (BSA) standard of linear range dilutions from 0.5 to 200 μg/mL. Assay was performed in triplicate for each sample. Samples chosen for analysis corresponded to 17 and 57 h in glucose, 48 and 72 h in xylose, and 26 and 44 h in acetate. Assay results represent cumulative proteome during each growth phase of yeast.

## Lipidomics

To quantify lipids and determine their fatty acid composition, quantitative gas chromatography–mass spectrometry (GC-MS) analysis with the internal standard method was used, similar as described in Tammekivi et al. 2019, [54]. Before analysis, cell pellets were lyophilized and derivatized by using acid-catalyzed methylation. This derivatization procedure produces methylated fatty acids from both free and bonded fatty acids. The quantitative analysis and derivatization procedure of the TAGs and free fatty acids was based on Tammekivi et al. 2021, [55]. From the lyophilized cells, 10–12 mg of was weighed into a 15 mL glass vial. An analytical balance (Precisa, Dietikon, Switzerland, resolution of 0.01 mg) was used to weigh all components that influence the quantitative analysis (samples, solvents, internal standard). To the cells, 2 mL of MeOH (≥99.9%, Honeywell, Charlotte, NC, USA) was added and the vial was sonicated for 15 min. Then, 0.4 mL of conc. $H_2SO_4$ (98%, VWR Chemicals, Radnor, PA, USA) was carefully added to the solution and the derivatization mixture was heated for 3 h at 80˚C. After, the mixture was extracted 3 x 2 mL with hexane (≥97.0%, Honeywell) and the extracts were pipetted through a layer of $K_2CO_3$ (99.5%, Reakhim) on top of a glass wool (Supelco,

Bellefonte, PA, United States) layer. The combined extracts were evaporated to dryness and the residue was redissolved in 2 mL of toluene ($\geq$99.9%, Honeywell). Depending on the expected fatty acid concentration, toluene and internal standard (hexadecane, $\geq$99%, Honeywell) were added so that the results would stay in the range of the calibration curve.

The solutions containing the fatty acid methyl esters and internal standard were analyzed with an Agilent (Santa Clara, CA, USA) 7890A GC connected to an Agilent 5975C inert XL mass spectrometric detector (MSD) with a triple-axis detector and an Agilent G4513A autosampler. The column was an Agilent DB-225MS capillary column (30 m x 0.25 mm diameter, 0.25 μm film thickness) with a (50%-cyanopropylphenyl)-methylpolysiloxane stationary phase. The temperatures of the mass spectrometer transfer line and ion source were 280˚C and 230˚C, respectively. The temperature of the inlet was 300˚C, injection volume 0.5 μL, and splitless mode was used, where the split was opened after 2 min. The oven's temperature program was the following: isothermal for 2 min at 80˚C, increased 20˚C/min to 200˚C, isothermal for 4 min, increased 5˚C/min to 220˚C, isothermal for 5 min, increased 10˚C/min to 230˚C, isothermal for 12 min. The total run time was 34 min. Electron ionization (EI) with 70 eV was used and the solvent delay was 5.6 min. Helium 6.0 was used as the carrier gas (flow rate 1.5 mL/min). Qualitative analysis was performed in the scan mode (mass range of 27–400 *m/z*) and quantitative analysis was performed in the selected ion monitoring (SIM) mode, which were both measured during the same GC-MS run. For data analysis, Agilent MSD ChemStation and NIST Mass Spectral Library Search 2.0 were used.

Commercial standard mixture of fatty acid methyl esters (FAME, C8–C24, Supelco) was used to confirm the identity (based on retention times, in addition to the mass spectral comparison) and to quantify the fatty acids. Seven calibration solutions were made from the FAME mixture in toluene and the same internal standard (hexadecane) was added. All calibration solutions were measured in random order in the same GC-MS sequence with the derivatized sample solutions. For each methylated fatty acid, a calibration curve was constructed based on the data obtained from the GC-MS analysis of the calibration solutions–$S_{AD}/S_{IS}$ vs. $C_{AD}/C_{IS}$–where S represents the peak area, C the concentration, AD the derivatized fatty acid, and IS the internal standard. Knowing the $S_{AD}/S_{IS}$ and $C_{IS}$ for the sample solution, if was possible the calculate the derivatized fatty acid concentration ($C_{AD}$). Finally, the obtained value was recalculated to represent the concentrations of particular fatty acids or homotriglycerides. Also the derivatization efficiency (for more information see Tammekivi et al. 2019, [51]) was taken into account by applying the same derivatization procedure and quantitative analysis for the analysis of five fatty acid standards (C16:0, C18:0, C18:1, C18:2 and C18:3) and their corresponding TAG standards. The obtained yield (% from the weighed quantity of the corresponding standard) was used to correct the result of the sample analysis. The sum of the quantified TAGs was presented as the total lipid content. Samples chosen for analysis corresponded to 24, 52 and 100 h in glucose, 48 and 96 h in xylose, and 39 and 84 h in acetate. Analysis results represent cumulative lipidome during each growth phase of the yeast.

## Experimental procedure for absolute proteomics

Absolute proteome quantification was performed using a nanoscale liquid chromatography with tandem mass spectrometry (Nano-LC/MS/MS), similar as described in Sanchez et al. 2021, [26]. Experimental procedure for cell lysis and sample preparation was done as described in the same study. Briefly, cell pellets were lysed using a pH 8.0 buffer (6 M guanidine HCl, 100 mM Tris-HCl, 20 mM dithiothreitol) and homogenized using the FastPrep-24 device (2x at 4 m/s for 30 s). After centrifugation and overnight precipitation (10% trichloroacetic acid, at 4˚C), protein concentration was measured as described above in the total protein content

section. For absolute quantification, proteome samples were mixed heavy-labeled *R. toruloides* grown in previously described minimal medium supplemented with heavy 15N, 13C-lysine (Silantes, Munich, Germany), which was used as an internal standard [1]. Further sample preparation and Nano-LC/MS/MS analysis was similar to previous descriptions [26]. Samples chosen for analysis corresponded to 17 and 57 h in glucose, 48 and 72 h in xylose, and 26 and 44 h in acetate, same as in the total proteome analysis.

## Proteomics data analysis

The raw data obtained from the Nano-LC/MS/MS analysis was processed using MaxQuant **v1.6.1.0** software package [56] with similar settings as described in Sanchez et al. 2021, [26]. Data search was performed against the Uniprot ([www.uniprot.org](www.uniprot.org)) *R. toruloides* NP11 reference proteome database [10]. Raw data quantification was similar to previous descriptions, except that the MS intensities were normalized with the average internal standard abundance (reverse Ratio H/L normalized). MS intensities were calculated from the internal standard abundance using the number of theoretically observable peptides (iBAQ, intensity Based Absolute Quantification; iBAQ H) feature in MaxQuant, the reverse Ratio H/L normalized of the sample, and reverse Ratio H/L. The resulting MS intensities were adjusted for 80% recovery of the sample injected. Finally, absolute protein concentrations were derived from the normalized sum of MS intensities assuming its proportionality to the measured total protein content, also known as the total protein approach [26].

LC-MS/MS data have been deposited to the ProteomeXchange Consortium ([http://proteomecentral.proteomexchange.org](http://proteomecentral.proteomexchange.org)) via the PRIDE partner repository [57] with the dataset identifier PXD037281. Processed quantitative data are presented in **S1 Dataset**. Duplicate experiments were used in differential expression analysis. *p*-values were adjusted for multiple comparisons using Benjamini-Hochberg (1995) method [58].

## Enzyme-constrained model reconstruction

Enzyme-constrained genome-scale metabolic model of *R. toruloides* was generated using the metabolic network *rhto-GEM* **version 1.3.0** [15]. The workflow was based on a semi-automatic algorithm of the GECKO toolbox **version 2.0.2** [17] operating on MATLAB (The MathWorks Inc., Natick, MA, United States). Model development was tracked on a dedicated Github repository: [https://github.com/alinarekena/ecRhtoGEM/](https://github.com/alinarekena/ecRhtoGEM/).

Firstly, functions *addMets*, *addGenesRaven* and *addRxns* from RAVEN [59] were used to add the alternative xylose assimilation pathway to rhto-GEM, as provided in ecRhtoGEM/ *edit_rhtoGEM*. Next, pipelines *enhanceGEM* and *generate_protModels* from the GECKO Toolbox were used to generate ec-models, as provided in ecRhtoGEM/ *reconstruct_ecRhtoGEM*.

During the *enhanceGEM* pipeline, enzyme kinetic parameters were relaxed to overcome model over constraint using the *manualModifications* function from the GECKO Toolbox. The enzymes subject to manual $k_{cat}$ value curation were identified by running *enhanceGEM* pipeline initially with the physiology data of the xylose condition, as provided in ecRhtoGEM/ customGECKO/*getModelParameters*. The *relative_proteomics.txt* and *uniprot.tab* input data were used to match enzymes with the model and retrieve their $k_{cat}$ values from the BRENDA database. The data for *uniprot.tab* were downloaded from Uniprot.org with *R. toruloides* strain NP11 as query, while *relative_proteomics.txt* contained average protein abundances of enzymes detected in our proteomics analysis (in mmol/gDCW). The *getModelParameters* function was used to ensure that the newly generated ec-model was constrained with experimental data. The GECKO Toolbox automatically performed the initial sensitivity analysis on

the objective function (ie. maximize cell growth) with respect to the individual $k_{cat}$ values by identifying the top limiting value and by iteratively replacing it with the maximum value available in BRENDA. According to the reported information, we adjusted $k_{cat}$ values identified as limiting to reasonably higher values found in literature (for a detailed description see ecRhtoGEM/*manualModifications*). In the next step, we used the *topUsedEnzymes* function from GECKO Toolbox to calculate the top ten enzyme usages in a mass-wise way (data not provided). Similarly as in the previous step, $k_{cat}$ values of enzymes identified among top used in each condition were increased to reasonably higher values referenced in the literature. In the script the procedure was named round A. Later, in so called round B, the *topUsedEnzymes* function was applied to the same conditions again and $k_{cat}$ values were modified until enzyme usage represented less than 1% of total protein pool, as provided in ecRhtoGEM/*manualModifications*. The final list of modified $k_{cat}$ values included 27 enzymes, as summarized in **S6 Table**. As automatic $k_{cat}$ values were derived from studies that involved not the same organism and substrate, their values were often very low. For example, the $k_{cat}$ value of fructose-bisphosphatase was increased from 0.002 $s^{-1}$ to 127 $s^{-1}$, on the basis of specific enzyme activity for the same EC number.

During the *generate_protModels* pipeline, growth- and non-growth-associated energy requirements were fit using measured substrate uptake and gas rates from batch cultivations of *R. toruloides* obtained in this study, as provided in ecRhtoGEM/customGECKO/fermentationData. They were set from 124.4 to 140.0 mmol/gDCW and from 0 to 3.65 mmol/(gDCW/h). Coefficients in oxidative phosphorylation from rhto-GEM were not changed. Polymerization costs from the study in *S. cerevisiae* [60] were used, similarly as in *rhto-GEM*. Average enzyme saturation factor (sigma) was fit to physiological parameters (ecRhtoGEM/results/enhanceGEM_pipeline/sigma), and set at 0.35 in ecRhtoGEM/customGECKO/getModelParameters. Biomass composition was modified from rhto-GEM to include *R. toruloides* CCT 7815 protein content, lipid content and acyl chain profiles, as provided in ecRhtoGEM/customGECKO and ecRhtoGEM/data, respectively. The *scaleLipidProtein* and *scaleLipidsRhto* functions from GECKO Toolbox and SLIMEr [61] were modified for the *generate_protModels* pipeline. To avoid the model to over constrain, automatic flexibilization was performed on concentrations of 7 (XNlim) to 25 (Gexp) enzymes, as listed in **S6 Table** (the old and new values are available at ecRhtoGEM/results/generate_prot_Models_pipeline/modifiedEnzymes.txt). An alternative approach to calculate the abundance of those enzymes for which no enzyme level had been measured was used as additional modification in addition to previously described modifications in the pipeline to handle the ow protein levels observed in Nlim conditions. In this approach, we directly subtracted the measured enzyme concentration (Pmeasured) from the total enzyme concentration (enzymeConc) to obtain the unmeasured enzyme concentration (PpoolMass). Modifications to original approach, by which GECKO adjusts for the unmeasured enzyme concentration, are available from *generate_protModels* and *constrainEnzymes* functions at 'customGECKO' folder. These included sample specific f-factor calculation was moved before filtering proteomics data (*generate_protModels*). This ensured higher coverage, while not largely affecting f calculation. Total protein content (Ptot) calculation was rescaled by adding standard deviation and flexibilization because of too low measurement. Then f, which was calculated in the beginning of *generate_protModels*, and rescaled updated Ptot were used to calculate expected total enzyme concentration (enzymeConc), as provided in *constrainEnzymes*. This ensured higher coverage, likely critical in low total protein content biomass (in case of all Nlim phases). Other updates included rescaling of enzyme usage to prevent very low fluxes, as provided in *generate_protModels*.

## Model calculations

Flux balance analysis was performed with the RAVEN toolbox using Gurobi solver (Gurobi Optimization Inc., Houston, Texas, United States). Flux variability was estimated with random sampling of the solution space with 2000 sampling iterations for each condition (ie., ec-model). For each sample, a random set of three reactions was given random weights and the sum of these were parsimoniously maximized to explore the constraint solution space [32], considering 1% variability from maximal growth rate and substrate uptake rate, 10% variability from predicted carbon dioxide production and oxygen consumption rate, 10% variability from measured by-product rates, 10% variability from protein pool, and 1% variability from NGAM, as specified in *analyze_ecRhtoGEM*. In glucose condition, simulated values were used to constrain gas exchange (carbon dioxide and oxygen) due to measurement problems in experimental values. In xylose condition, measured values were used to constrain the production of by-products xylitol and D-arabinitol. To allow the model to use either traditional or alternative xylose assimilation pathway, xylulokinase (XK) was not blocked, but eventually constrained with enzyme constraints from the protein pool. In acetate condition, measured values were used to constrain the production of citrate. Flux value was calculated as a median of 2000 sampling iterations. Flux variability was represented as SD divided by flux, multiplied by 100. Finally, fluxes were converted to base GEM formalism using *mapRxnsToOriginal* function from Domenzain et al. 2022 [30]. For the analysis, fluxes were normalized by dividing absolute flux with the specific substrate uptake rate to ensure the comparability among different conditions. Additional data analysis was performed on ATP, NADPH and NADH turnover extracted using the *getMetProduction* function from [7]. Yield was calculated as turnover (sum of fluxes) divided by the specific rate of substrate uptake. Apparent catalytic activities ($k_{app}$, $s^{-1}$) were calculated according to Eq (2).

$$k_{app} = \frac{flux}{E} \tag{2}$$

Where flux refers to median flux, mmol/gDCW/h, obtained from 2000 iterations of random sampling of the solution space [32] and E refers to mean protein concentration (n = 2), mmol/gDCW.

## Supporting information

**S1 Table. Physiological characterization parameters in *R. toruloides* CCT 7815 batch cultivations on three different carbon substrates—glucose (G) (63 g/L, C/N 68.6), xylose (X) (70g/L, C/N 80) and acetate (A) (20 g/L, C/N 80) at exponential growth (exp) and nitrogen limitation (Nlim) phases.**
(XLSX)

**S2 Table. Gene and metabolite names of *R. toruloides* selected for annotation in Figs 2–4 in main text.**
(XLSX)

**S3 Table. Using absolute proteomic data to calculate translation rate in *R. toruloides* batch cultivations on three different carbon substrates–glucose (G), xylose (X) and acetate (A)—during exponential growth (exp) and nitrogen limitation (Nlim) phases.**
(XLSX)

**S4 Table. Proteins whose concentration and the $k_{cat}$ value were integrated in the enzyme-constrained models of *R. toruloides* representing batch cultivations on three different carbon substrates–glucose (G), xylose (X) and acetate (A)—during exponential growth (exp)**

and nitrogen limitation (Nlim) phases. Proteins required EC numbers to allow the algorithm to query their $k_{cat}$ values, therefore the existing and new EC numbers were provided to the input file (uniprot.tab) for the GECKO algorithm (columns J-K). In case of multiple EC numbers found for the same gene in rhto-GEM, EC numbers were combined. For further details on how the algorithm selected the $k_{cat}$ values based on their EC numbers, see [17].
(XLSX)

**S5 Table. Enzymatic reactions constrained with enzyme abundances in enzyme-constrained genome-scale models of *R. toruloides* for batch cultivations on three different carbon substrates–glucose (G), xylose (X) and acetate (A)—during exponential growth (exp) and nitrogen limitation (Nlim) phases.**
(XLSX)

**S6 Table. Enzymes with flexibilized concentrations and/or their $k_{cat}$ values for the enzyme-constrained genome-scale metabolic models of *R. toruloides* for batch cultivations on three different carbon substrates–glucose (G), xylose (X) and acetate (A)—during exponential growth (exp) and nitrogen limitation (Nlim) phases.** Flexibilization of the measured enzyme abundances was performed automatically by the algorithm in the GECKO Toolbox. Flexibilization of $k_{cat}$ values was performed manually by changing the $k_{cat}$ values retrieved automatically from BRENDA, based on suggestions by the algorithm. Detailed information on flexibilized protein concentrations at ecRhtoGEM repository /results/generate_protModels_pipeline. Detailed information on modified $k_{cat}$ values at ecRhtoGEM repository /customGECKO/manualModifications.
(XLSX)

**S1 Dataset. MS intensities (arbitrary unit) and absolute protein concentrations (µg/g_protein) in *R. toruloides* batch cultivations on three different carbon substrates–glucose (G), xylose (X) and acetate (A)–during exponential growth (exp) and nitrogen limitation (Nlim) phases.** Absolute concentrations are calculated using total protein amount (TPA) quantification method of duplicate conditions. Normalization refers to 80% recovery of the sample injected applied to the sum of intensities.
(XLSX)

**S2 Dataset. Absolute protein abundances (µg/g_protein) in *R. toruloides* batch cultivations on three different carbon substrates–glucose (G), xylose (X) and acetate (A)–during exponential growth (exp) and nitrogen limitation (Nlim) phases.** Concentrations are calculated using total protein amount (TPA) quantification method. Pairs having adjusted p-value < 0.05 and log2 fold change (log2FC) > |1| of average of duplicate conditions were considered significantly differentially expressed. P value was adjusted for multiple comparisons (n = 3100) using Benjamini & Hochberg method [58]. Protein abundances were filtered by excluding instances, where standard deviation exceeds mean value of two replicates. Normalization refers to 80% recovery of the sample injected applied to the sum of intensities.
(XLSX)

**S3 Dataset. Enzyme turnover numbers ($k_{cat}$, $s^{-1}$) and apparent catalytic activities ($k_{app}$, $s^{-1}$) of *R. toruloides* in batch cultivations on three different carbon sources–glucose (G), xylose (X) and acetate (A)—during exponential growth (exp) and nitrogen limitation (Nlim) phases.** $k_{cat}$ values were retrieved from BRENDA using the GECKO Toolbox [17]. $k_{app}$ values were obtained by dividing flux, mmol/gDCW/h, by protein abundance, mmol/gDCW. Flux refers to median from 2000 iterations of random sampling of the solution space [32].

Normalization of protein abundance refers to 80% recovery of sample injected.
(XLSX)

**S4 Dataset. Flux predictions in *R. toruloides* batch cultivations on three different carbon substrates–glucose (G), xylose (X) and acetate (A) at exponential growth (exp) and nitrogen limitation (Nlim) phases.** Fluxes are calculated using random sampling of the solution space with 2000 iterations (mmol/gDCW/h) on *R. toruloides* enzyme-constrained genome-scale models. Fluxes represent median values and are normalized by dividing flux with specific substrate uptake rate (representing % of carbon distribution). Fluxes are represented in non-ec model (base GEM) annotation by merging forward and reverse fluxes created by the GECKO formalism. Flux variability is SD divided by the flux value, multiplied by 100. Flux changes were compared using log2 fold change (log2FC).
(XLSX)

**S5 Dataset. Flux predictions with acetate kinase added (t_0886) (phosphate transacetylase removed, t_0082) in *R. toruloides* batch cultivations on three different carbon substrates–glucose (G), xylose (X) and acetate (A) at exponential growth (exp) and nitrogen limitation (Nlim) phases.** Fluxes are calculated using random sampling of the solution space with 2000 iterations (mmol/gDCW/h) on *R. toruloides* enzyme-constrained genome-scale models. Fluxes represent median values and are normalized by dividing flux with specific substrate uptake rate (representing % of carbon distribution). Fluxes are represented in non-ec model (base GEM) annotation by merging forward and reverse fluxes created by the GECKO formalism. Flux variability refers to SD divided by the flux value, multiplied by 100.
(XLSX)

**S6 Dataset. Flux predictions with blocked phosphoketolase (t_0081) in *R. toruloides* batch cultivations on three different carbon substrates–glucose (G), xylose (X) and acetate (A) at exponential growth (exp) and nitrogen limitation (Nlim) phases.** Fluxes are calculated using random sampling of the solution space with 2000 iterations (mmol/gDCW/h) on *R. toruloides* enzyme-constrained genome-scale models. Fluxes represent median values and are normalized by dividing flux with specific substrate uptake rate (representing % of carbon distribution). Fluxes are represented in non-ec model (base GEM) annotation by merging forward and reverse fluxes created by the GECKO formalism. Flux variability refers to SD divided by the flux value, multiplied by 100.
(XLSX)

**S7 Dataset. Flux predictions with NAD/NADH as cofactor for DAD-2/LXR (t_0884) in *R. toruloides* batch cultivations on xylose- (X) based chemically defined medium at exponential growth (exp) and nitrogen limitation (Nlim) phases.** Fluxes are calculated using random sampling of the solution space with 2000 iterations (mmol/gDCW/h) on *R. toruloides* enzyme-constrained genome-scale models. Fluxes represent median values and are normalized by dividing flux with specific substrate uptake rate (representing % of carbon distribution). Fluxes are represented in non-ec model (base GEM) annotation by merging forward and reverse fluxes created by the GECKO formalism. Flux variability refers to SD divided by the flux value, multiplied by 100.
(XLSX)

**S1 Fig. Growth curves of batch cultivation of *R. toruloides* CCT 7815 on three different carbon substrates at nitrogen limitation.** (A) glucose (63 g/L, C/N 68.6), (B) xylose (70g/L, C/N 80) and (C) acetate (20 g/L, C/N 80). Arrows in red are used to denote sampling points for proteomics and protein content measurements. Average of duplicate experiments with SD in

extracellular metabolites concentration (g/L) and intracellular lipid content (g_lipid/gDCW) is illustrated. Curves represent a single measurement in bioreactor 2 (R2) in $CO_2$ (%), specific growth rate μ ($h^{-1}$) and biomass concentration (g/L), while for the rate calculations used for modelling duplicate conditions were used.
(TIF)

**S2 Fig. D-arabinitol detection in supernatant obtained from *R. toruloides* batch cultivations in xylose-based chemically defined medium (70 g/L).** Figures represent HPLC profiles of D-arabinitol during nitrogen limitation phase on xylose (XNlim) performed at 20˚C. Column: Chiralpak; eluent: hexane-ethanol (70,30, v/v). Flow rate 0.3 mL/min; detection: refractive index.
(TIF)

**S3 Fig. Venn diagrams of significantly differentially expressed proteins in *R. toruloides* during batch cultivations on three different carbon substrates–glucose (63 g/L), xylose (70 g/L) and acetate (20 g/L)–under nitrogen limitation conditions.** (A) Comparison between exponential growth (exp) and nitrogen limitation (Nlim) phase. (B) Comparison among substrates during exp phase. Comparison was made using μg/g of total protein. Pairs having adjusted p-value < 0.05 and log2 fold change > |1| were considered significantly differentially expressed. P value was adjusted for multiple comparisons (n = 3100) according to Benjamini & Hochberg (1995).
(TIF)

**S4 Fig. Proteome integration into enzyme-constrained models for *R. toruloides* in batch cultivations on three different carbon substrates–glucose (G), xylose (X) and acetate (A)—during exponential growth (exp) and nitrogen limitation (Nlim) phases.** (A) Protein count as searched against the reference proteome database of *R. toruloides* strain NP11. (B) Mass-wise coverage of proteome in models (g_protein/g_DCW).
(TIF)

**S5 Fig. Apparent enzyme catalytic activities, $k_{app}$, 1/s, of *R. toruloides* in batch cultivations on three different carbon sources–glucose (G), xylose (X) and acetate (A)—during exponential growth (exp) and nitrogen limitation (Nlim) phases.** $k_{app}$ calculated using fluxes from flux balance analysis on enzyme-constrained models of *R. toruloides* and measured enzyme absolute abundances. Frequency of $k_{app}$ values represented in log10 scale.
(TIF)

**S6 Fig. Flux predictions in *R. toruloides* batch cultivations on three different carbon substrates–glucose (G), xylose (X) and acetate (A) at exponential growth (exp) and nitrogen limitation (Nlim) phases.** Fluxes are calculated using random sampling of the solution space with 2000 iterations (mmol/gDCW/h) on *R. toruloides* enzyme-constrained genome-scale models. Fluxes represent median values and are normalized by dividing flux with specific substrate uptake rate (representing % of carbon distribution). PPP: pentose phosphate pathway; TCA cycle: tricarboxylic acid cycle. Gene names and corresponding metabolic reaction IDs are included in S2 Table.
(TIF)

**S7 Fig. Fluxes carrying NADPH in *R. toruloides* on glucose- (G) based chemically defined medium at exponential growth (exp) (A) and nitrogen limitation (Nlim) (B) phase (mmol/gDCW/h).** Fluxes are calculated using random sampling of the solution space with 2000 iterations (mmol/gDCW/h) on *R. toruloides* enzyme-constrained genome-scale models. Fluxes represent median values from flux sampling. Negative fluxes denote metabolite consumption,

positive fluxes denote metabolite production. Percentage (%) denotes the flux divided by NADPH turnover (sum of absolute fluxes involving NADPH). Gene names and corresponding metabolic reaction IDs are included in S2 Table.
(TIF)

**S8 Fig. Predicted ATP turnover (mmol/gDCW/h) (A) and ATP yield (mmol_ATP/ mmol_substrate) (B) in *R. toruloides* on three different carbon substrates–glucose (G), xylose (X) and acetate (A)–in a chemically defined medium at exponential growth (exp) and nitrogen limitation (Nlim) phases.** ATP turnover is calculated as a sum of fluxes involving ATP. ATP yield is calculated as turnover divided by specific rate of substrate uptake. Fluxes are predicted using random sampling of the solution space with 2000 iterations (mmol/ gDCW/h) on *R. toruloides* enzyme-constrained genome-scale models. Median flux values are used in calculations.
(TIF)

**S9 Fig. Predicted NADH turnover (mmol/gDCW/h) (A) and NADH yield (mmol_NADH/ mmol_substrate) (B) in *R. toruloides* on three different carbon substrates—glucose (G), xylose (X) and acetate (A)–in a chemically defined medium at exponential growth (exp) and nitrogen limitation (Nlim) phases.** NADH turnover is calculated as sum of absolute fluxes involving NADH. NADH yield is calculated as turnover divided by specific rate of substrate uptake. Fluxes are predicted using random sampling of the solution space with 2000 iterations (mmol/gDCW/h) on *R. toruloides* enzyme-constrained genome-scale models. Median flux values are used in calculations.
(TIF)

**S10 Fig. Average absolute enzyme abundances (µg/g_protein) in *R. toruloides* batch cultivations on three different carbon substrates–glucose (G), xylose (X) and acetate (A)–during exponential growth (exp) and nitrogen limitation (Nlim) phases.** Absolute enzyme concentrations are calculated using total protein amount (TPA) quantification method. Results of duplicate experiments with SD are represented. PPP: pentose phosphate pathway; TCA cycle: tricarboxylic acid cycle. Full names of gene abbreviations are included in S2 Table.
(TIF)

**S11 Fig. Apparent enzyme catalytic activities, $k_{app}$, $s^{-1}$, of *R. toruloides* in batch cultivations on three different carbon sources–glucose (G), xylose (X) and acetate (A)—during exponential growth (exp) and nitrogen limitation (Nlim) phases.** $k_{app}$ calculated using fluxes from flux balance analysis on enzyme-constrained genome-scale models of *R. toruloides* and measured enzyme absolute abundances. PPP: pentose phosphate pathway; TCA cycle: tricarboxylic acid cycle. Full names of gene abbreviations are included in S2 Table.
(TIF)

**S12 Fig. Fluxes carrying NADPH in *R. toruloides* on xylose- (X) based chemically defined medium at exponential growth (exp) (A) and nitrogen limitation (Nlim) (B) phase (mmol/ gDCW/h).** Fluxes are calculated using random sampling of the solution space with 2000 iterations (mmol/gDCW/h) on *R. toruloides* enzyme-constrained genome-scale models. Fluxes represent median values from flux sampling. DAD-2/LXR is considered NADP-dependent. Negative fluxes denote metabolite consumption, positive fluxes denote metabolite production. Gene names and corresponding metabolic reaction IDs are included in S2 Table.
(TIF)

**S13 Fig. Predicted NADPH turnover (mmol/gDCW/h) (A) and NADPH yield (mmol_ NADPH/mmol_substrate) (B) in *R. toruloides* on three different carbon substrates—**

**glucose (G), xylose (X) and acetate (A)–in a chemically defined medium at exponential growth (exp) and nitrogen limitation (Nlim) phases.** NADPH turnover is calculated as sum of absolute fluxes involving NADPH. NADPH yield is calculated as turnover divided by specific rate of substrate uptake. Fluxes are predicted using random sampling of the solution space with 2000 iterations (mmol/gDCW/h) on *R. toruloides* enzyme-constrained genome-scale models. Median flux values are used in calculations.
(TIF)

**S14 Fig. Fluxes carrying NADH in *R. toruloides* on xylose- (X) based chemically defined medium at exponential (exp) (A) and nitrogen limitation (Nlim) (B) phase (mmol/gDCW/h).** Fluxes are calculated using random sampling of the solution space with 2000 iterations (mmol/gDCW/h) on *R. toruloides* enzyme-constrained genome-scale models. Fluxes represent median values from flux sampling. DAD-2/LXR is considered NADP-dependent. Negative flux denotes metabolite consumption, positive flux denotes metabolite production. Letters [m] and [c] denote compartments of cytoplasm and mitochondria. Gene names and corresponding metabolic reaction IDs are included in S2 Table.
(TIF)

**S15 Fig.** Fluxes carrying ATP in *R. toruloides* on acetate-(A) based chemically-defined medium at exponential growth (exp) (A) and nitrogen limitation (Nlim) (B) phases (mmol/gDCW/h). Fluxes are calculated using random sampling of the solution space with 2000 iterations (mmol/gDCW/h) on *R. toruloides* enzyme-constrained genome-scale models. Fluxes represent median values from flux sampling. Negative flux denotes metabolite consumption, positive flux denotes metabolite production. Letters [m] and [c] denote compartments of cytoplasm and mitochondria. Gene names and corresponding metabolic reaction IDs are included in S2 Table.
(TIF)

**S16 Fig. Fluxes carrying NADH in *R. toruloides* on acetate- (A) based chemically defined medium at exponential growth (exp) (A) and nitrogen limitation (Nlim) (B) phase (mmol/gDCW/h).** Fluxes are calculated using random sampling of the solution space with 2000 iterations (mmol/gDCW/h) on *R. toruloides* enzyme-constrained genome-scale models. Fluxes represent median values from flux sampling. Negative flux denotes metabolite consumption, positive flux denotes metabolite production. Letters [m] and [c] denote compartments of cytoplasm and mitochondria. Gene names and corresponding metabolic reaction IDs are included in S2 Table.
(TIF)

**S17 Fig. Fluxes carrying NADPH in *R. toruloides* on acetate- (A) based chemically defined medium at exponential growth (exp) (A) and nitrogen limitation (Nlim) (B) phase (mmol/gDCW/h).** Fluxes are calculated using random sampling of the solution space with 2000 iterations (mmol/gDCW/h) on *R. toruloides* enzyme-constrained genome-scale models. Fluxes represent median values from flux sampling. Negative flux denotes metabolite consumption, positive flux denotes metabolite production. Gene names and corresponding metabolic reaction IDs are included in S2 Table.
(TIF)

**S18 Fig. D-arabinitol enantiomer detection using HPLC analysis.** Figure represents retention times for arabinitol separation in Chiralpak column, at 20˚C, hexane-ethanol (70:30, v/v) mixture.
(TIF)

## Acknowledgments

The authors thank Laura Kibena for HPLC analysis, Proteomics Core Facility at University of Tartu for proteome quantification. Lipid analysis was carried out using the instrumentation at the Estonian Center of Analytical Chemistry (www.akki.ee).

## Author Contributions

**Conceptualization:** Nemailla Bonturi, Petri-Jaan Lahtvee.

**Data curation:** Alīna Reķēna.

**Funding acquisition:** Petri-Jaan Lahtvee.

**Investigation:** Alīna Reķēna, Marina J. Pinheiro, Eliise Tammekivi, Petri-Jaan Lahtvee.

**Methodology:** Alīna Reķēna, Eliise Tammekivi, Eduard J. Kerkhoven.

**Project administration:** Petri-Jaan Lahtvee.

**Supervision:** Nemailla Bonturi, Isma Belouah, Koit Herodes, Eduard J. Kerkhoven, Petri-Jaan Lahtvee.

**Visualization:** Alīna Reķēna.

**Writing – original draft:** Alīna Reķēna.

**Writing – review & editing:** Alīna Reķēna, Marina J. Pinheiro, Nemailla Bonturi, Isma Belouah, Eliise Tammekivi, Koit Herodes, Eduard J. Kerkhoven, Petri-Jaan Lahtvee.

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
