## [Decision Letter · Decision Letter 0]

14 Dec 2022

Dear Prof Lahtvee,

Thank you very much for submitting your manuscript "Genome-scale metabolic modeling reveals metabolic trade-offs associated with lipid production in Rhodotorula toruloides" for consideration at PLOS Computational Biology.

As with all papers reviewed by the journal, your manuscript was reviewed by members of the editorial board and by several independent reviewers. In light of the reviews (below this email), we would like to invite the resubmission of a significantly-revised version that takes into account the reviewers' comments.

Please review carefully the comments from reviewer 1 especially around the estimation of the kinetic parameters for this non-model organism and the use of flux sampling for the estimation.

We cannot make any decision about publication until we have seen the revised manuscript and your response to the reviewers' comments. Your revised manuscript is also likely to be sent to reviewers for further evaluation.

Sincerely,

Radhakrishnan Mahadevan

Guest Editor

PLOS Computational Biology

Mark Alber

Section Editor

PLOS Computational Biology

Please review carefully the comments from reviewer 1 especially around the estimation of the kinetic parameters for this non-model organism and the use of flux sampling for the estimation.

Reviewer's Responses to Questions

**Comments to the Authors:**

Reviewer #1: In this work, the authors measured extracellular fluxes and absolute protein concentrations under six conditions of varying substrate types and availability. Data for N-limited conditions were generated for R. toruloides, which are valuable resources for the research community since N-limitation is typically used to induce lipid overproduction. Genome-scale metabolic model was used as a central platform to integrate the flux and proteomic data and provide insights into intracellular fluxes and metabolic responses to growth condition changes. The authors also provide many valuable discussions using their results and published papers. However, the author needs to re-consider how the kinetic parameters were estimated and flux modeling was done. In addition, there are many minor points that the authors need to clarify. This manuscript is highly recommended for publication, once the following below concerns are addressed.

Major concerns

1. Page 4, line 73-75, the authors need to also mention the ATP cost to convert acetyl-CoA to malonyl-CoA. This also appears again on page 16, line 359.

2. Page 5, line 93-98, can the authors provide a comment (or perhaps a more explicitly statement) on if xylulokinase is not encoded in the genome, or if it is encoded but not expressed?

3. Page 7, line 134-136, this sentence contradicts a later statement which is that separate versions of the model were built for each growth condition.

4. Page 7, line 154, it would be helpful to list all “parameters” here as an overview for what to be discussed.

5. Description of “mineral medium” is unclear. What are in “1 mL/L vitamin solution”? What are the differences compared to the commonly used Yeast Nitrogen Base (YNB) media for yeasts?

6. Page 9, line 196-201, the two sentences contrast each other. First sentence mentions similarities whereas the second sentence mentions differences between strains. The authors need to re-write and add more explanations to these sentences to prevent confusion. “The final fatty acid compositions were in agreement with the previous data reported on glucose and xylose [8]. In our study, the growth phase-dependent differences in compositions were linked to PUFAs, mainly linoleate (C18:2), while in the previous study [8] it was linked mostly to palmitate (C16:0), which possibly reflect the fact that different R. toruloides strains were used in these studies.”

7. Page 9, line 203 (plus any other instances of the same type), in the unit of biomass yield “gDCW/g_carbon”, what does “g_carbon” refer to? Is this “g_substrate” (e.g., carbon substrate of glucose)? Using “g_substrate” is better since “g_carbon” might mislead the readers for only the carbon content of substrate.

8. Page 10, the author mentioned that “ However, we were able to measure only 68.5% of carbon during the Nlim phase (S1 Table). Likely, it was due to the fact that R. toruloides strain CCT 7815 was making cell aggregates in the minimal mineral glucose-based medium.”. This is a very important observation and lesson for future researchers in working with R. toruloides. Could the authors further elaborate how “making cell aggregates” lead to incomplete mass balance in measurements? This might be unclear to readers, especially those without extensive experience working with the organism experimentally.

The authors then moved on to mention that: “The growth curves were highly similar when using ammonium sulfate or urea (S1A Figure), and therefore, further analysis with glucose was carried out with cells using urea as a nitrogen source.” The authors need to make it clearer for what results correspond to what nitrogen source (i.e., ammonium and urea). This includes: (i) exp or N-lim phase, (ii) glucose only or not (which is already clear), (iii) applied for which data (i.e., on Fig. 1 and for proteomics and/or flux data), and (iv) recorded in Supplementary Materials.

9. Line 241-245, PCA has also been done for R. toruloides using transcriptomics data (Jagtap et al., 2021, https://doi.org/10.1007/s00253-021-11549-8). It would be informative if the authors can provide few sentences comparing results of PCA using proteomics vs. transcriptomics. Do similar patterns arise in both? Are there any noticeable differences?

10. In page 15, where and how are the kcat values obtained for R. toruloides, given that this is a non-model organism whose reaction kinetic parameter data is not readily available? This information is important to be (at least briefly) mentioned in the main text so that the readers have a perspective on what (organism/data) sources are enzymatic efficiency parameters taken from.

11. In page 15, the following sentence does not make sense: “Convex Basis sampling algorithm [28] with 2000 sampling points and a random set of objective functions to be maximized to evaluate flux variability”. Why would sampling of a solution space require an optimization objective?

12. Page 16, line 341, “apparent” (or “effective”) should be used instead of “actual” to describe kapp to be consistent with previous publications. Line 345, do the authors mean “in vivo” instead of “in silico” when describing the kapp?

13. Page 16, line 343-344, the authors set model parameters as follow: “kapp is set by the in vitro enzyme turnover number kcat”. This might omit kapp > kcat cases being valid because of the in vivo enzyme activity enhancement effect. There are examples on in vivo kapp (calculate using similar methods to that used in this manuscript) available in the literature for the authors to read.

14. In calculating kapp, the authors should consider using parsimonious FBA solution to represent minimally required flux activity instead of using flux sampling. This is because sampling will generate noise which mean that the kapp calculated from sampled flux is unreliable for low flux pathways, such as for fatty acid oxidation as noted by the authors: “Some of the lowest kapp values in acetate condition were associated with fatty acid degradation and beta oxidation metabolic pathways.”. Fatty acid degradation function under these conditions is not clear and thus their fluxes calculated by flux sampling are unreliable anyway.

15. In page 17, line 372, the authors need to provide evidence from their own data or from the literature on phosphoketolase pathway activity. At least, the authors should comment and follow up on: (i) enzyme expression (for all of the steps), (ii) FBA simulation without phosphoketolase being active to confirm no alternative pathways exist to functionally replace it, and (iii) definite resolution of phosphoketolase pathway would require a follow-up metabolic flux analysis study (using isotopically labeled tracer). This is necessary so as not to mislead readers on being confident about phosphoketolase pathway activity.

In page 18, the authors comment on the ATP citrate lyase pathway to be a functional equivalent to phosphoketolase in producing cytosolic acetyl-CoA. Why is this pathway not being active instead of phosphoketolase? This might be resolved if the authors perform simulation with phosphoketolase or ATP citrate lyase being inactive and analyze the results.

16. Page 21, line 443-445, it is not clear why this is mentioned: “However, protein levels of L-xylulose reductase (EC 1.1.1.10, RHTO_00373) and L-arabinitol dehydrogenase (EC 1.1.1.12, RHTO_01629) were also 10-fold upregulated during growth on xylose versus other substrates.”

17. Page 22, line 459-461, the authors mentioned: “In support of this mechanism, the metabolic model could only predict a feasible solution of LXR/DAD-2 only using NADP(+) as the cofactor.” Was this simulation performed with enzyme constraint (with kcat or kapp < kcat being active)? The authors need to also perform simulation with FBA (i.e., without fluxes constrained by enzyme levels) to assess if infeasibility is due to metabolic demand or due to enzyme level constraining flux. The authors might want to re-calculate and use a new kapp value if simulation with FBA is indeed feasible.

18. Page 24, line 512-513, isn’t the decrease in ACC flux simply due to the fact that lipid still need to be produced even in exp phase? This means that the lipid produced for the much higher growth rate in exp phase is higher in absolute amount compared to that in N-lim phase.

19. Page 24, line 526, is this ME the cytosolic one? This needs to be clarified since there is also mitochondrial ME.

20. Page 26, line 563, what are the other “alternative enzymes of the nitrogen assimilation pathway”? This needs to be clarified since GDH is commonly known to be the only nitrogen fixation pathways with ammonium as nitrogen substrate. Also, wasn’t urea used instead of ammonium as mentioned at a previous point? Later on, line 568-570, is this a typo “GDH1 catalyzes the final steps in the conversion of urea to glutamate and glutamine, which serve as the sources of cellular nitrogen”?

21. Page 30, line 649-653, the authors bring up a confusing correlation between carbon flux entering TCA cycle and malate-aspartate shuttle. Malate-aspartate shuttle net result is the transport of NADH from cytosol to mitochondria, which means carbon is not transported (as a net result). The authors need to elaborate, or this statement needs to be removed or rewritten for clarity.

22. Page 32, line 706-711, I see that the authors are discussing kcat values here. This statement needs to be placed at a point before mentioning modeling results so that readers can have correct interpretations of the results. Is the kcat used the highest value among multiple values for multiple organisms? If this is the case, should kapp be preferred even if its value exceed kcat value?

23. Were C/N ratios accounted for nitrogens in both ammonium and urea? On page 34, line 748-750, how exactly were the C/N ratios controlled? At what time points was C/N ratio control applied (e.g., end of exponential phase which is before N-lim phase)? Was C/N ratio value controlled by continuous or fed-batch mode?

24. Page 43, if no proteomic data is available, should the reactions not be constrained by enzyme level instead of using made-up enzyme levels? This can probably be done by setting a very high kcat (e.g., perhaps 1E18) in GECKO to make the constraint artificially inactive.

Minor concerns

1. Page 4, line 71, change “Origins of” to “Metabolic pathways producing”

2. Page 5, line 101, change “enzymatic” to “biochemical”, as GEM also contains non-enzymatic transport, exchange, and spontaneous reactions.

3. Page 7, line 160-161, for better understanding, the authors need to rephrase the following: “induction of hydrolysate-tolerance and lipid accumulation genes without physiological changes regarding growth and substrate consumption”

4. Page 8, line 161-162, “cultures were grown at a 162 molar C/N ratio from 69 to 80” should be rephrased to indicate C/N ratio being specific to provided substrate (as indicated in the Methods).

7. In Figure 1’s footer, the authors need to explain the substrates denoted as “G”, “X”, and “A” (e.g., glucose is denoted as “G” ). It would be better if figure 1C is split into two separate bar charts for growth rates and glucose uptake rates.

8. The authors need to improve the English language usage in the manuscript for better text comprehension.

Reviewer #2: Recommendation: minor revision

In this article, the authors collected physiological parameters (including the lipid synthesis rate, cell growth rate, cell consumption, by-product synthesis, and lipid composition) for the growth phase and the lipid synthesis phase of R. toruloides in different carbon sources. These experimental data were integrated into Genomic metabolic network. The metabolic model a gave a detailed interpretation of how flux patterns are changing in R. toruloides on different substrates during the exponential growth and in lipid accumulation. This work contributes to an improvement in knowledge of R. toruloides metabolism. I recommend to accept it before addressing the following issues.

1. Please give the full name of “P/O ratio” when it appears at the first time.

2. Line 84, No XPK-related content was found in reference 10. Please double check.

3. Line 214-217, Is there any reference to support or explain the fact that the 68.5% of the carbon detected is due to cell aggregates? Why cell aggregates influence it?

4. Line 222, Please correct the spelling in the coordinate units of figure 1b and 1d. What is the color stands for in figure 1c, 1e and 1f. Please make it clear.

5. Many supplement figures don’t have a detailed figure legend. Please give more details. Readers should understand the figure based on its legend.

**Have the authors made all data and (if applicable) computational code underlying the findings in their manuscript fully available?**

Reviewer #1: **No: **The github page mentioned in the manuscript is inaccessible https://github.com/alinarekena/ecRhtoGEM

Reviewer #2: Yes

PLOS authors have the option to publish the peer review history of their article (what does this mean?). If published, this will include your full peer review and any attached files.

Reviewer #1: No

Reviewer #2: No
---

## [Decision Letter · Decision Letter 1]

7 Mar 2023

Dear Prof Lahtvee,

We are pleased to inform you that your manuscript 'Genome-scale metabolic modeling reveals metabolic trade-offs associated with lipid production in Rhodotorula toruloides' has been provisionally accepted for publication in PLOS Computational Biology.

Best regards,

Radhakrishnan Mahadevan

Guest Editor

PLOS Computational Biology

Mark Alber

Section Editor

PLOS Computational Biology

Reviewer's Responses to Questions

**Comments to the Authors:**

Reviewer #1: The authors have addressed all concerns from the reviewer. The manuscript is recommended to be accepted for publication.

Reviewer #2: I recommend to accept this paper

**Have the authors made all data and (if applicable) computational code underlying the findings in their manuscript fully available?**

Reviewer #1: Yes

Reviewer #2: Yes

PLOS authors have the option to publish the peer review history of their article (what does this mean?). If published, this will include your full peer review and any attached files.

Reviewer #1: No

Reviewer #2: No

---

## [Editor Report · Acceptance letter]

21 Apr 2023

PCOMPBIOL-D-22-01517R1 

Genome-scale metabolic modeling reveals metabolic trade-offs associated with lipid production in *Rhodotorula toruloides*

Dear Dr Lahtvee,

I am pleased to inform you that your manuscript has been formally accepted for publication in PLOS Computational Biology. Your manuscript is now with our production department and you will be notified of the publication date in due course.

With kind regards,

Zsofia Freund
